# ATF4-dependent fructolysis fuels growth of glioblastoma multiforme

Chao Chen[1,5], Zhenxing Zhang[1,5], Caiyun Liu[1,2], Bin Wang[1,2], Ping Liu[1], Shu Fang[1,2], Fan Yang[1,2], Yongping You[3,4] & Xinjian Li [1,2] ✉

Excessive consumption of fructose in the Western diet contributes to cancer development. However, it is still unclear how cancer cells coordinate glucose and fructose metabolism during tumor malignant progression. We demonstrate here that glioblastoma multiforme (GBM) cells switch their energy supply from glycolysis to fructolysis in response to glucose deprivation. Mechanistically, glucose deprivation induces expression of two essential fructolytic proteins GLUT5 and ALDOB through selectively activating translation of activating transcription factor 4 (ATF4). Functionally, genetic or pharmacological disruption of ATF4-dependent fructolysis significantly inhibits growth and colony formation of GBM cells in vitro and GBM growth in vivo. In addition, ATF4, GLUT5, and ALDOB levels positively correlate with each other in GBM specimens and are poor prognostic indicators in GBM patients. This work highlights ATF4-dependent fructolysis as a metabolic feature and a potential therapeutic target for GBM.

Glioblastoma multiforme (GBM) is a highly malignant primary brain tumor with a very poor prognosis. Approaches for treating GBM are limited due to the lack of understanding for the initiation and progression of this disease[1]. Metabolic reprogramming, a core hallmark of cancer, contributes to both the initiation and progression of many cancers[2]. For GBM, it primarily metabolizes glucose for energy production even in the presence of ample oxygen, a phenomenon known as the Warburg Effect[3]. To overcome the tumor microenvironment of low glucose levels, GBM cells utilize other energetic substrates, such as fructose, amino acids, and fatty acids, to support rapid proliferation of cancer cells[3,4]. Among these substrates, fructose is the second most abundant blood sugar in humans, with a normal physiological serum concentration ranging from 0.5 to 1 mM[5,6].

Fructose is a monosaccharide derived from the diet. The uptake of fructose is mediated by a high-affinity fructose transporter GLUT5, which is encoded by the *SLC2A5* gene, in an insulin-independent manner[7]. Intracellular fructose is phosphorylated by ketohexokinase (KHK) to generate fructose 1-phosphate (F1P) using adenosine

triphosphate (ATP) as a phosphate donor. F1P is split by aldolase B (ALDOB) to form dihydroxyacetone phosphate (DHAP) and glyceraldehyde (GA). These two intermediate metabolites enter the later stages of glycolysis after the GA is converted to glyceraldehyde 3-phosphate (GAP) by triokinase[7]. Fructose metabolism, therefore, completely bypasses the first and second glycolytic regulatory steps, leading to greater lipogenesis compared with lipid production resulting from glycolysis[8–10]. Although fructose metabolism, under normal physiological conditions, has been extensively investigated, it is still unknown how fructose metabolism impacts GBM tumorigenesis.

In response to diverse stress stimuli, phosphorylation of the eukaryotic translation initiation factor 2 α subunit (eIF2α) on serine 51 represses global protein synthesis. Coincidentally, this inhibition of eIF2α stimulates selective translation of activating transcription factor 4 (ATF4), a master transcription factor, which activates a common adaptive pathway termed the Integrate Stress Response (ISR)[11,12]. In mammals, four eIF2α kinases, including PKR-like endoplasmic reticulum kinase (PERK), general control non-depressible 2 (GCN2) kinase,

[1]CAS Key Laboratory of Infection and Immunity, CAS Center for Excellence in Biomacromolecules, Institute of Biophysics, Chinese Academy of Sciences, Beijing 100101, China. [2]College of Life Sciences, University of Chinese Academy of Sciences, Beijing 100049, China. [3]Department of Neurosurgery, The First Affiliated Hospital of Nanjing Medical University, Nanjing 210029, China. [4]Institute for Brain Tumors, Jiangsu Key Lab of Cancer Biomarkers, Prevention and Treatment, Jiangsu Collaborative Innovation Center for Cancer Personalized Medicine, Nanjing Medical University, Nanjing 211166, China. [5]These authors contributed equally: Chao Chen, Zhenxing Zhang. ✉e-mail: lixinjian@ibp.ac.cn

protein kinase R (PKR), and heme-regulated inhibitor (HRI) kinase, have been identified as activators of the ISR. These kinases are known to be activated by endoplasmic reticulum (ER) stress, amino acid starvation, viral infection, and heme deprivation[13].

A highly active glucose metabolism has been observed in GBM cells[3]. The increased glycolytic flux would accelerate glucose consumption, resulting in the stress of glucose deprivation in the GBM microenvironment. However, how GBM cells adapt to and overcome this environmental stress is unclear. In the present study, we demonstrate that ATF4-dependent fructolysis supports GBM progression under glucose-deprived conditions. Transcription factor ATF4, activated by glucose deprivation, induces expression of fructolytic genes, leading to energy supply switching from glycolysis to fructoslysis in GBM cells undergoing glucose deprivation. Pharmacological blockage of fructose utilization inhibits the GBM growth, showing a therapeutic potential for GBM.

## Results

### Glucose deprivation induces fructolysis

To determine whether glucose deprivation affects fructose consumption, we treated the GBM cell lines including U87, LN229, and A172, genetically characterized primary GBM cells TJ46 (Supplementary Table 1), as well as GBM stem cell line GSC23 with glucose-deprived media for 18 hours. Fructose metabolic rate assays demonstrated that glucose deprivation markedly increased rates of fructose metabolism (Fig. 1a). Consistent with glucose deprivation, treatment with the glucose metabolism inhibitor 2-deoxy-D-glucose (2-DG), but not with sorbitol, increased rates of fructose metabolism in a 2-DG dose-dependent manner (Supplementary Fig. 1a). These results were supported by fructose consumption analyses showing that glucose deprivation (Fig. 1b) and 2-DG treatment (Supplementary Fig. 1b) increased fructose consumption. Quantitative PCR (qPCR) and

immunoblot analyses revealed that glucose deprivation-induced mRNA (Fig. 1c) and protein (Fig. 1d) expression of the fructolytic genes including *SLC2A5* and *ALDOB*, but not *KHK* and *TKFC*. Consistent results were obtained by treating GBM cells with 2-DG in a dose-dependent manner (Supplementary Fig. 1c, d). These results imply glucose deprivation promotes fructolysis of GBM cells.

We next titrated the extracellular glucose concentration needed to activate fructose metabolism by treating the U87 and LN229 cells with media containing different glucose concentrations and found that medium glucose levels lower than 1.2 mM are able to significantly upregulate fructolytic rate (Supplementary Fig. 1e) and induce expression of GLUT5 and ALDOB (Supplementary Fig. 1f). Consistently, upregulation of fructolytic rate (Supplementary Fig. 1g) and induced expression of GLUT5 and ALDOB (Supplementary Fig. 1h) resulted from glucose deprivation was abrogated by treating U87 and LN229 cells with media containing glucose levels of more than 1.2 mM. These results suggest that fructose metabolism in GBM cells is activated when extracellular glucose concentration is <1.2 mM. Moreover, immuno-blotting analyses demonstrated that uH2B (K120 mono-ubiquitination of histone H2B), a semiquantitative histone marker for examining glucose levels that the tumor cells are exposed to[14], was undetectable in U87 and LN229 cells treated with media containing glucose levels of <1.2 mM (Supplementary Fig. 1f, h), suggesting that uH2B may be used as a marker for examining glucose deprivation in tumor tissues.

### ATF4 is required for glucose deprivation-induced fructolysis

It has been shown that glucose deprivation triggered both the unfolded protein response (UPR) and integrated stress response (ISR), depending on activation of the protein kinases PERK or GCN2, respectively, both of which phosphorylate eIF2α[15–17]. Treatment of U87 and LN229 cells with the PERK inhibitor GSK2656157 or the GCN2 inhibitor A-92 partially blocked glucose-deprivation-induced mRNA (Fig. 2a) and protein (Fig. 2b) expression of *SLC2A5* and *ALDOB*. Furthermore, combined treatment with GSK2656157 and A-92 completely blocked glucose-deprivation-induced expression of these genes (Fig. 2a, b). However, these effects were not observed when the U87 and LN229 cells were treated with compound C (Fig. 2a, b), an inhibitor of another glucose-deprivation-activated protein kinase, namely 5′ AMP-activated protein kinase (AMPK)[18]. Therefore, these data strongly suggest that glucose deprivation induces expression of fructolytic genes depending on PERK-eIF2α and GCN2-eIF2α, but not AMPK signaling pathways.

Phosphorylation of eIF2α results in selective translation of the transcription factor ATF4[11]. To determine whether ATF4 is required for glucose-deprivation-induced expression of fructolytic genes, we knocked out *ATF4* in U87, LN229, and A172 cells. As expected, glucose deprivation-induced protein (Fig. 2c) and mRNA (Fig. 2d) expression of *SLC2A5* and *ALDOB*, but not *KHK* and *TKFC*, whereas glucose-deprivation-induced expression of *SLC2A5* and *ALDOB* was abrogated by *ATF4* knockout (KO) (Fig. 2c, d). In contrast, exogenous expression of ATF4 markedly upregulated the protein (Supplementary Fig. 2a) and mRNA (Supplementary Fig. 2b) expression of *SLC2A5* and *ALDOB*, but not *KHK* and *TKFC*, suggesting that *KHK* and *TKFC* are expressed in GBM cells in an ATF4-independent manner. Functionally, *ATF4* KO abolished the increase of fructose metabolic rate (Fig. 2e) and fructose consumption (Fig. 2f) induced by glucose deprivation. Conversely, exogenous expression of ATF4 enhanced the fructose metabolic rate (Supplementary Fig. 2c) and fructose consumption (Supplementary Fig. 2d). Taken together, these data demonstrate that glucose deprivation induces fructolysis of GBM cells in an ATF4-dependent manner.

### ATF4 binds to promoters of fructolytic genes to activate fructolysis upon glucose deprivation

It is known that ATF4 activates its target gene expression by binding to a motif containing C/EBP-ATF Response Element (CARE) sequence (5′-TGATGXAAX-3′)[19]. To determine the global genomic regulation by

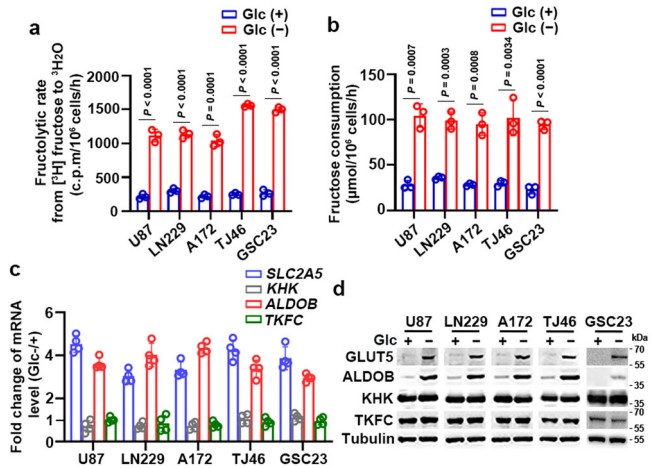

**Fig. 1 | Glucose deprivation promotes fructolysis. a** U87, LN229, A172, TJ46, and GSC23 cells were deprived of glucose for 18 h. The fructose metabolic rate was measured by monitoring the conversion of D-[5-³H] fructose to ³H₂O and normalized to cell number. c.p.m, counts per minute. **b** U87, LN229, A172, TJ46, and GSC23 cells were deprived of glucose for 18 hours and then incubated with 10 mM of fructose for another 18 hours. The media were collected for analysis of fructose consumption. **c, d** U87, LN229, A172, TJ46, and GSC23 cells deprived of glucose for 18 hours were analyzed by quantitative PCR (**c**) and immunoblotting with indicated antibodies (**d**). Data were normalized with β-actin mRNA levels and presented as fold change induced by glucose deprivation (**c**). Glc (+) and Glc (−) represent glucose-supplemented (25 mM glucose) and -deprived (1 mM glucose) condition, respectively (**a**–**d**). Data represent the mean ± SD of three (**a**, **b**) or four (**c**) independent experiments. The experiments were repeated three times independently with similar results (**d**). *P* values were determined by the two-tailed Student's *t*-test (**a**, **b**). Source data are provided as a Source Data file.

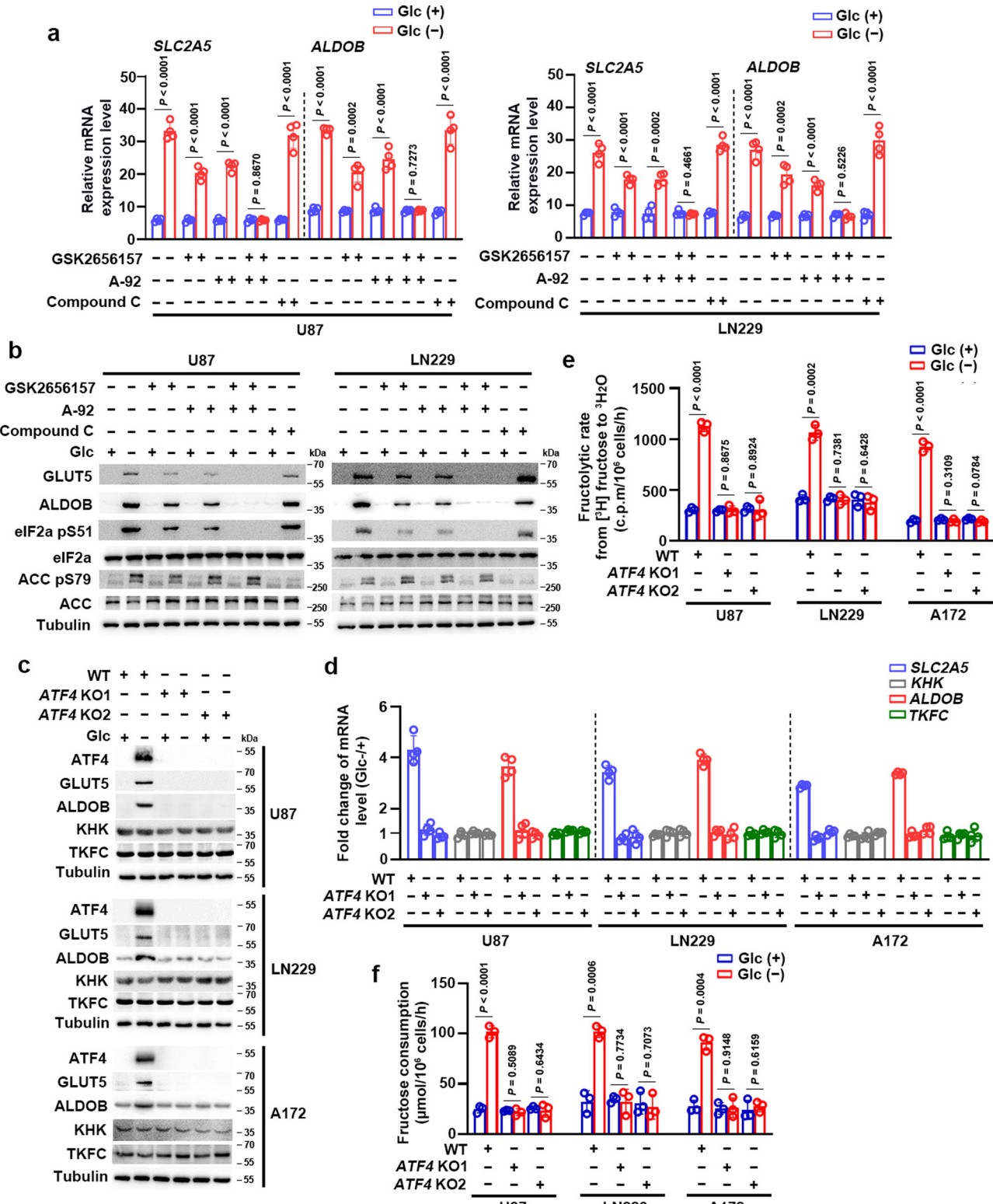

**Fig. 2 | ATF4 is required for glucose deprivation-induced fructolysis. a, b** U87 and LN229 cells treated without or with glucose deprivation for 18 hours in the absence or presence of indicated inhibitors were analyzed by quantitative PCR (**a**) and immunoblotting with indicated antibodies (**b**). Data were normalized with β-actin mRNA levels and presented as relative mRNA expression level (**a**). **c, d** U87, LN229 and A172 cells without or with *ATF4* knockout (KO) were deprived of glucose for 18 h and then analyzed by immunoblotting with indicated antibodies (**c**) and quantitative PCR (**d**). Data were normalized with β-actin mRNA levels and presented as fold change induced by glucose deprivation (**d**). **e** U87, LN229, and A172 cells without or with *ATF4* KO were deprived of glucose for 18 hours. The fructose

metabolic rate was measured by monitoring the conversion of D-[5-³H] fructose to ³H₂O and normalized to cell number. c.p.m, counts per minute; WT, wild type. **f** U87, LN229 and A172 cells without or with *ATF4* KO were deprived of glucose for 18 h and then incubated with 10 mM of fructose for another 18 h. The media were collected for analysis of fructose consumption. Data represent the mean ± SD of three (**e, f**) or four (**a, d**) independent experiments. The experiments were repeated three times independently with similar results (**b, c**). *P* values were determined by the two-tailed Student's *t*-test (**a, e, f**). Source data are provided as a Source Data file.

ATF4, we performed chromatin immunoprecipitation (ChIP) combined with high-throughput sequencing (ChIP-seq) experiments using an anti-ATF4 antibody. Indeed, ChIP-seq analyses demonstrated that signals from ATF4 ChIP were enriched at transcription start site (TSS) regions (±1 kb) upon glucose deprivation (Supplementary Fig. 3a and Supplementary Data 1) and that CARE is a de novo motif recognized by ATF4 (Supplementary Fig. 3b). Notably, we found that the genes related to fructolysis were significantly enriched in the ATF4 ChIP peaks (Supplementary Fig. 3c). ATF4 bound to CARE motifs located within promoters of fructolytic genes including *SLC2A5* and *ALDOB*, but not *KHK* and *TKFC*, upon glucose deprivation (Fig. 3a). Furthermore, we mutated the CARE motifs located within promoters of *SLC2A5* or *ALDOB* via CRISPR-Cas9-mediated knockin (Supplementary Fig. 3d). ChIP analysis demonstrated that glucose deprivation-induced ATF4 binding at wild-type (WT) CARE motifs, but not mutated CARE motifs, located within promoters of *SLC2A5* and *ALDOB* (Supplementary Fig. 3e).

To check whether CARE motifs located within promoters of *SLC2A5* and *ALDOB* are essential for the expression of these genes in response to glucose deprivation, we inserted the promoters of *SLC2A5* and *ALDOB* into a luciferase reporter plasmid. Notably, a luciferase assay demonstrated that disruption of CARE motifs by point mutation impaired the glucose-deprivation-induced transcription activity of *SLC2A5* and *ALDOB* promoters (Fig. 3b). Consistently, GBM cells with disruption of CARE motifs located within the promoters of *SLC2A5* or *ALDOB* (Supplementary Fig. 3d) failed to induce mRNA (Fig. 3c) and protein (Fig. 3d) expression of these genes, or increase fructose metabolic rate (Fig. 3e) and fructose consumption (Fig. 3f) upon glucose deprivation. Collectively, these results suggest that ATF4 induces fructolysis through binding to CARE motifs located within the promoters of fructolytic genes *SLC2A5* or *ALDOB*.

### ATF4-dependent fructolysis rescues proliferation and colony formation of GBM cells under glucose-deprived condition

To determine the role of ATF4-dependent fructolysis, we examined the proliferation and colony formation of GBM cells without or with *ATF4* KO (Fig. 2c) or ATF4 binding deficiency in the promoters of *SLC2A5* or *ALDOB* (Supplementary Fig. 3d) under high glucose (25 mM), normal glucose (6 mM) and glucose-deprived conditions supplemented without or with fructose. As expected, fructose supplementation rescued the inhibition of cell growth (Fig. 4a, b) and colony formation (Fig. 4c, d) induced by glucose deprivation. These effects were abrogated by *ATF4* KO (Fig. 4a, c) or disruption of ATF4 binding motifs in the promoters of *SLC2A5* or *ALDOB* (Fig. 4b, d). Intriguingly, fructose supplementation did not affect cell growth and colony formation of these GBM cells cultured under high and normal glucose conditions (Supplementary Fig. 4a–d). These results suggest that ATF4-dependent fructolysis is required to maintain growth and colony formation of GBM cells under glucose-deprived conditions. In addition, we observed that A172 cells, but not U87, LN229 and TJ46 cells, are resistant to cell death induced by severe glucose deprivation (Supplementary Fig. 4e), which is consistent to the previous report[20]. These data imply that fructolysis supports the proliferation of GBM cells regardless of their resistance to cell death under glucose-deprived conditions.

We next generated the *SLC2A5-* or *ALDOB*-knockdown U87 cells by infecting U87 cells with lentiviruses expressing shRNA targeting *SLC2A5* or *ALDOB* (Supplementary Fig. 4f). As expected, *SLC2A5* or *ALDOB* knockdown impaired the fructose-mediated colony formation of U87 cells under glucose-deprived condition (Supplementary Fig. 4g). Furthermore, exogenous expression of GLUT5 and ALDOB rescued the inhibition of fructose-mediated colony formation in ATF4-knockdown U87 cells under glucose-deprived condition (Supplementary Fig. 4h, i). These data suggest that GLUT5 and ALDOB are required for fructose-mediated GBM cell survival under glucose-deprived condition.

### ATF4-dependent fructolysis is required to maintain GBM growth

To investigate the role of ATF4-dependent fructolysis in GBM growth, we intracranially injected luciferase-expressing U87 cells without or with *ATF4* KO (Fig. 4e) or ATF4 binding-deficiency in the promoters of *SLC2A5* or *ALDOB* (Fig. 4f) into athymic nude mice. Bioluminescent imaging analyses indicated that tumors with *ATF4* KO (Fig. 4e) or ATF4 binding deficiency in the promoters of *SLC2A5* or *ALDOB* (Fig. 4f) exhibited a significant decrease of tumor size on day 21, but not on day 3, post tumor-cell injection, consistently, mice bearing these tumors also exhibited an obviously prolonged survival time (Fig. 4g, h). These data suggest that ATF4-dependent fructolysis is instrumental during tumor progression, but not tumor initiation. Similarly, orthotopic tumorigenesis analyses demonstrated that exogenous expression of GLUT5 and ALDOB (Supplementary Fig. 4h) markedly eliminated the *ATF4*-knockdown-mediated impairment of GBM growth (Supplementary Fig. 5a). Furthermore, we found that fructose administration promoted growth of tumors derived from U87 cells, but this effect was not observed in tumors derived from U87 cells with knockdown of *SLC2A5* or *ALDOB* (Supplementary Figs. 4f, 5b), suggesting that fructose-promoted GBM growth depends on GLUT5 and ALDOB.

Immunohistochemical (IHC) analyses with anti-ATF4, anti-GLUT5, anti-ALDOB, anti-Ki67, anti-cleaved PARP1 (cPARP1), and anti-uH2B antibodies demonstrated that tumor tissues derived from U87 cells with *ATF4* KO exhibited no expression of ATF4, low expression of GLUT5 and ALDOB, decreased expression of Ki67, and increased PARP1 cleavage (Supplementary Fig. 5c). Similar results were observed in tumor tissues derived from U87 cells with disruption of ATF4 binding motifs in the promoters of *SLC2A5* or *ALDOB* (Supplementary Fig. 5d). Notably, around 95% of tumor cells exhibited negative staining of uH2B on the tumor sections of each treatment group (Supplementary Fig. 5e, f), evidencing that a portion (~95%) of U87 cells in the tissues underwent glucose deprivation. In addition, we detected serum fructose in athymic nude mice fasted for 12 h. Mice-bearing tumors with *ATF4* KO (Supplementary Fig. 5g) or with disruption of ATF4 binding motifs (Supplementary Fig. 5h) exhibited increased serum fructose levels, suggesting that fructose utilization was reduced in these mice. In summary, these data demonstrate that ATF4-dependent fructolysis supports GBM growth in an orthotopic mouse model of GBM.

### Pharmacological blockage of fructose utilization suppresses proliferation and colony formation of GBM cells under glucose-deprived condition

Given that the fructose supplement rescues the decreased GBM cell proliferation resulted from glucose deprivation, we hypothesized that inhibition of fructose utilization with a pharmacological agent would be able to reduce GBM cell proliferation. We used a GLUT5 inhibitor[21], namely 2,5-anhydro-D-mannitol (2,5-AM), to block TJ46 and U87 cell's fructose utilization induced by glucose deprivation. Data demonstrated that 2,5-AM treatment markedly suppressed fructose-induced cell proliferation (Fig. 5a) and colony formation (Fig. 5b) under glucose-deprived condition, whereas 2,5-AM treatment had little influence on the glucose-induced cell proliferation (Supplementary Fig. 6a) and colony formation (Supplementary Fig. 6b). Furthermore, we observed that 2,5-AM treatment impaired the fructose-mediated colony formation of U87 cells under glucose-deprived condition, but this effect was not observed in U87 cells with knockdown of *SLC2A5* or *ALDOB* (Supplementary Figs. 4f, 6c), suggesting that 2,5-AM treatment inhibits GBM cell survival depending on GLUT5 and ALDOB expression. Together, these results demonstrate that 2,5-AM acts as a potent agent for inhibiting proliferation of GBM cells grown under glucose-deprived condition supplemented with fructose.

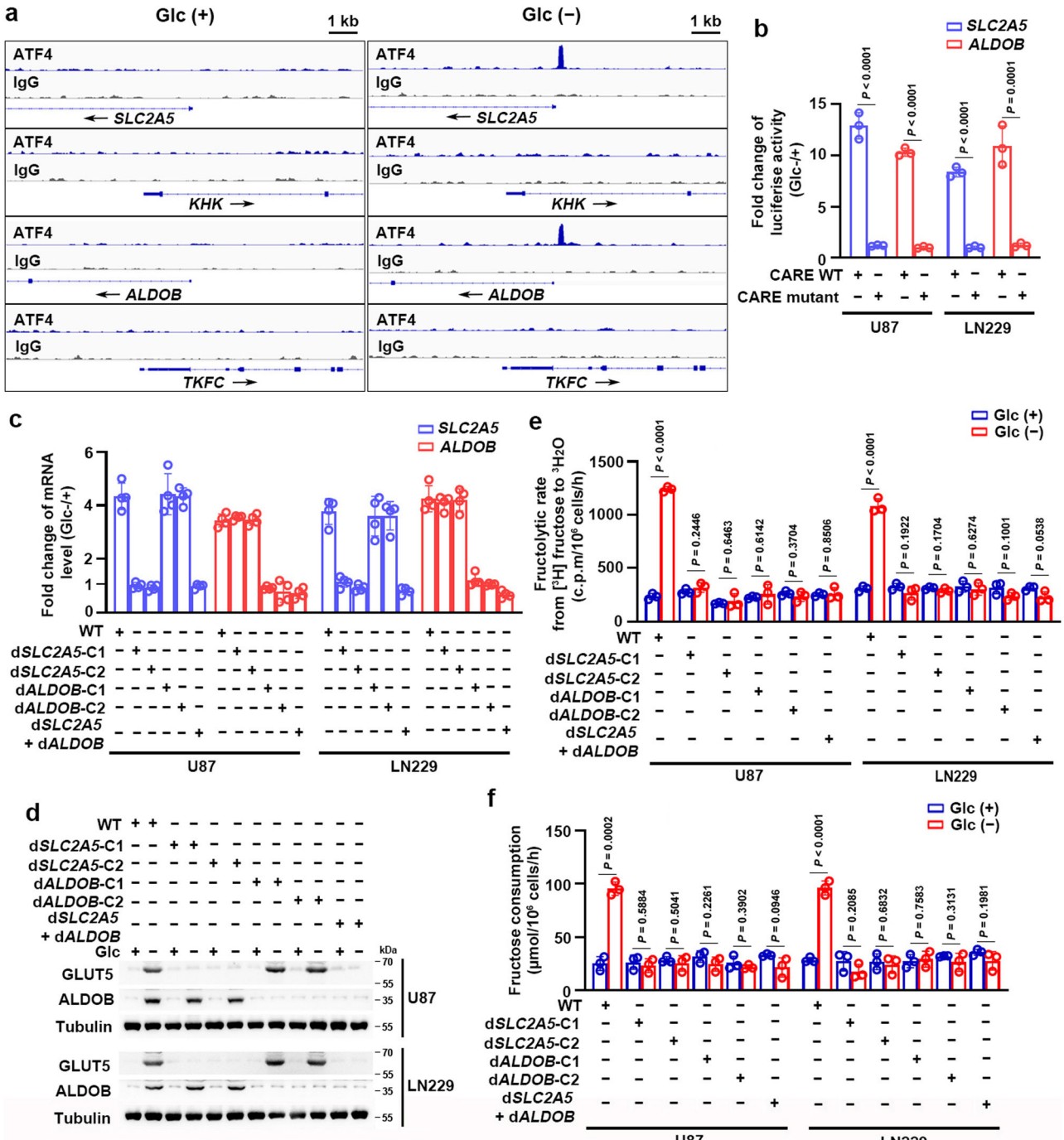

**Fig. 3 | ATF4 binds to the promoters of fructolytic genes to activate fructolysis upon glucose deprivation. a** U87 cells were treated without or with glucose deprivation for 18 h. Chromatin immunoprecipitation (ChIP) combined with high-throughput sequencing (ChIP-seq) was performed using the ATF4 antibody and the promoters of indicated genes were visualized using Integrated Genome Browser. This experiment was performed twice independently with similar results. **b** U87 cells were infected with luciferase-reporter lentiviruses carrying *SLC2A5* (−593 to +255 bp) or *ALDOB* (−661 to +214 bp) promoter without or with disruption of the CARE motif. These cells were treated without or with glucose deprivation for 18 h. The firefly luciferase activity was normalized with the cell numbers obtained from the duplicated samples and presented as fold change induced by glucose deprivation. CARE, C/EBP-ATF response element. **c**–**f** U87 and LN229 cells without or with disruption of the CARE motifs located within the promoters of *SLC2A5* or *ALDOB* were treated without or with glucose deprivation for 18 h. **c**, **d** Quantitative PCR (**c**) and immunoblotting with indicated antibodies (**d**) were performed. Data were normalized with β-actin mRNA levels and presented as fold change induced by glucose deprivation (**c**). **e** Indicated GBM cells were treated without or with glucose deprivation for 18 h. Fructose metabolic rate was measured by monitoring the conversion of D-[5-³H] fructose to ³H₂O and normalized to cell number. c.p.m, counts per minute. **f** Indicated GBM cells treated without or with glucose deprivation for 18 h were incubated with 10 mM of fructose for another 18 h. The media were collected for analysis of fructose consumption. WT, wild type; d*SLC2A5*-C1/2 and d*ALDOB*-C1/2, cell clone 1/2 with disruption of the CARE motifs located within the promoters of *SLC2A5* and *ALDOB*, respectively (**c**–**f**). Data represent the mean ± SD of three (**b**, **e**, **f**) or four (**c**) independent experiments. The experiments were repeated three times independently with similar results (**d**). *P* values were determined by the two-tailed Student's *t*-test (**e**, **f**). Source data are provided as a Source Data file.

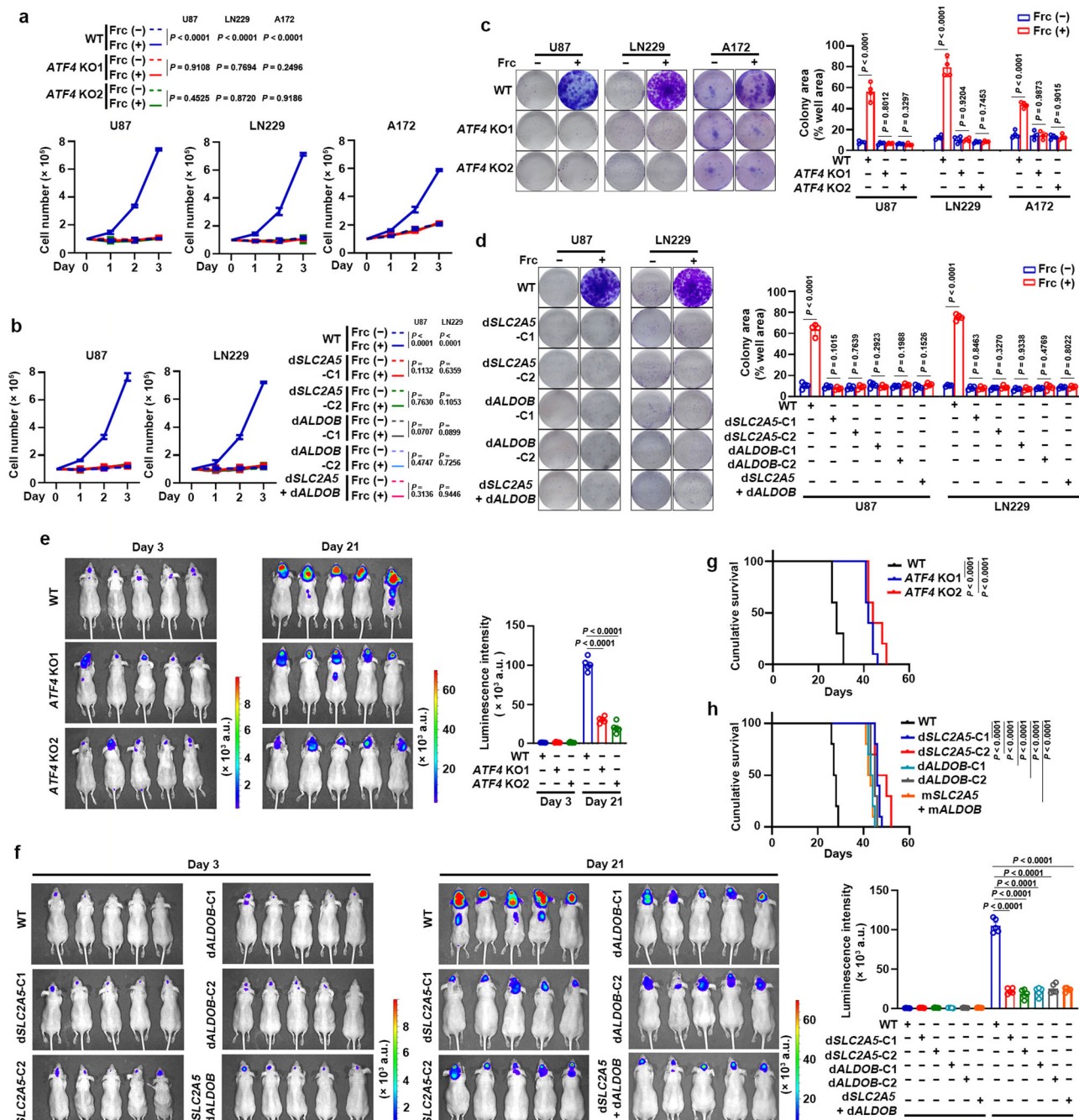

**Fig. 4 | ATF4-dependent fructolysis rescues glucose-deprivation impaired proliferation and colony formation of GBM cells and is required to maintain GBM growth. a, b** Indicated GBM cells without or with *ATF4* KO (**a**) or ATF4 binding deficiency in the promoters of *SLC2A5* or *ALDOB* (**b**) were cultured under glucose-deprived condition supplemented without or with fructose for 3 days. The cells were then collected and counted. **c, d** Indicated GBM cells without or with *ATF4* KO (**c**) or ATF4 binding deficiency in the promoters of *SLC2A5* or *ALDOB* (**d**) were cultured under glucose-deprived condition supplemented without or with fructose for 14 days. The cells were then fixed by 4% paraformaldehyde and stained with crystal violet. Colony formation areas in each dish were analyzed by ImageJ software. **e, f** Luciferase-expressing U87 cells without or with *ATF4* KO (**e**) or ATF4 binding deficiency in the promoters of *SLC2A5* or *ALDOB* (**f**) were intracranially injected into athymic nude mice (*n* = 5). Luminescence intensity derived from

tumors was measured on day 3 and day 21 post tumor-cell injection. Relative luminescence intensity was shown. a.u., arbitrary unit. **g, h** Luciferase-expressing U87 cells without or with *ATF4* KO (**g**) or ATF4 binding deficiency in the promoters of *SLC2A5* or *ALDOB* (**h**) were intracranially injected into athymic nude mice (*n* = 5). The survival times of the mice were recorded. Frc (−) and Frc (+) represent without and with 10 mM fructose supplementation, respectively (**a–d**). WT, wild type; d*SLC2A5*-C1/2 and d*ALDOB*-C1/2, cell clone 1/2 with disruption of the CARE motifs located within the promoters of *SLC2A5* and *ALDOB*, respectively (**b, d, f, h**). Data represent the mean ± SD of six (**a, b**) or four (**c, d**) independent experiments and the mean ± SEM of 5 mice (**e, f**). *P* values were determined by the two-tailed Student's *t*-test (**a–d**), one-way ANOVA (**e, f**), or two-tailed log-rank test (**g, h**). Source data are provided as a Source Data file.

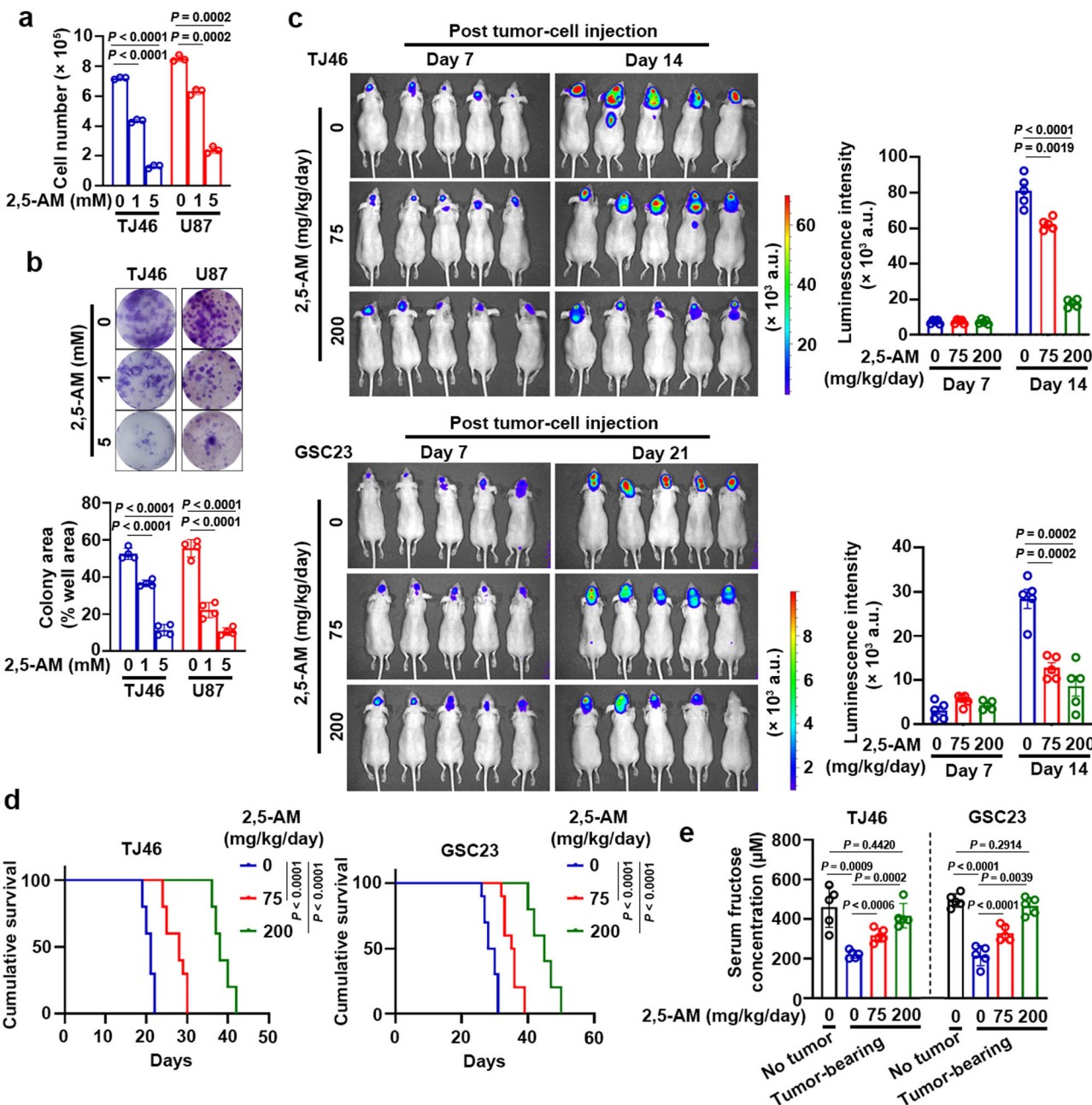

**Fig. 5 | Pharmacological blockage of fructose utilization suppresses fructose-induced proliferation and colony formation of GBM cells in vitro and shows therapeutic potential in vivo. a, b** TJ46 and U87 cells cultured under glucose-deprived condition supplemented with 10 mM of fructose were treated with the indicated concentration of fructose analog 2,5-AM for 3 (**a**) or 14 (**b**) days, and then collected and counted (**a**) or fixed by 4% paraformaldehyde and stained with crystal violet for colony formation analysis (**b**). **c** Luciferase-expressing TJ46 and GSC23 cells were intracranially injected into athymic nude mice (*n* = 5). Seven days after tumor-cell injection, 200 μl of 2,5-AM (75 or 200 mg kg⁻¹) or vehicle (PBS) was delivered to the mice via intraperitoneal administration daily for 7 days for TJ46 cells or 14 days for GSC23 cells. Luminescence intensity derived from tumors was measured at indicated time points post tumor-cell injection and relative luminescence intensity is shown. a.u., arbitrary unit. **d** Luciferase-expressing TJ46 and

GSC23 cells were intracranially injected into athymic nude mice (*n* = 5). Seven days after tumor-cell injection, 200 μl of 2,5-AM (75 or 200 mg kg⁻¹) or vehicle (PBS) was delivered to the mice via intraperitoneal administration daily. The survival times of the mice were recorded. **e** Luciferase-expressing TJ46 and GSC23 cells were intracranially injected into athymic nude mice (*n* = 5). Seven days after tumor-cell injection, indicated dosages of 2,5-AM were delivered to the mice via intraperitoneal administration daily. On day 14 (for TJ46 cells) or day 21 (for GSC23 cells) post tumor-cell injection, the mice were fasted for 12 h and then serum fructose concentration was determined by a quantitative colorimetric assay. The mice (*n* = 5) without injection of tumor cells served as no tumor control. Data represent the mean ± SD of six (**a**) or four (**b**) independent experiments and the mean ± SEM of 5 mice (**c**, **e**). *P* values were determined by the one-way ANOVA (**a–c**, **e**) or two-tailed log-rank test (**d**). Source data are provided as a Source Data file.

## Pharmacological blockage of fructose utilization shows therapeutic potential

To investigate the therapeutic potential of pharmacological blockage of fructose utilization in vivo, we intracranially injected the luciferase-expressing TJ46 and GSC23 cells into athymic nude mice. Four mouse groups were enrolled for each cell line, including a no tumor control

group, tumor-bearing mice treated with vehicle (vehicle group), and tumor-bearing mice treated with 2,5-AM at low (low 2,5-AM) or high (high 2,5-AM group) dosage. As expected, bioluminescent imaging indicated that administration of 2,5-AM into tumor-bearing mice significantly suppressed the brain tumor growth of mice in a dose-dependent manner (Fig. 5c). In line with this result, the use of 2,5-AM

prolonged the overall survival of tumor-bearing mice according to 2,5-AM concentration (Fig. 5d). In addition, 2,5-AM treatment inhibited growth of tumors derived from U87 cells, but this effect was not observed in tumors derived from U87 cells with knockdown of *SLC2A5* or *ALDOB* (Supplementary Figs. 4f, 6d). These data support that 2,5-AM treatment inhibits GBM growth depending on GLUT5 and ALDOB expression.

Next, IHC analyses with anti-ATF4, anti-Ki67, anti-cPARP1, and anti-uH2B antibodies demonstrated that 2,5-AM decreased expression of Ki67, increased caspase-mediated PARP1 cleavage, but did not affect the expression of ATF4 (Supplementary Fig. 6e). Intriguingly, around 40% of TJ46-tumor cells and 80% of GSC23-tumor cells displayed negative staining of uH2B on the tumor sections of each treatment group (Supplementary Fig. 6f), evidencing that a considerable portion of tumor cells in the tissues underwent glucose deprivation. In addition, compared with no tumor controls, the tumor-bearing mice exhibited reduced serum fructose levels (Fig. 5e), and administration of 2,5-AM significantly upregulated serum fructose concentrations in a 2,5-AM concentration-dependent manner (Fig. 5e), suggesting that serum fructose consumption resulted from tumorigenesis is suppressed by 2,5-AM treatment. Collectively, these data demonstrate that 2,5-AM is a potential agent that possesses antitumor activity against GBM.

### The expression levels of ATF4, GLUT5, and ALDOB positively correlate with each other in GBM specimens and indicate a poor prognosis in GBM patients

To determine the clinical relevance of our findings, we analyzed 68 human GBM specimens with antibodies against ATF4, GLUT5, and ALDOB (Fig. 6a). We revealed that the levels of ATF4, GLUT5, and ALDOB expression correlated with each other in GBM specimens (Fig. 6b). Next, we analyzed the survival durations in 68 GBM patients, all of whom underwent standard therapies. These data demonstrated that high levels of ATF4, GLUT5, and ALDOB expression in GBM specimens indicated a poor overall survival (Fig. 6c). Furthermore, multivariate analyses with adjustments for age, sex, and total resection status, all of which are relevant clinical covariates, indicated that high levels of ATF4, GLUT5, and ALDOB expression were independent prognostic factors for overall survival of GBM patients (Supplementary Table 2). Thus, these results indicate that ATF4-dependent fructolysis is associated with clinical aggressiveness of human GBM.

### Discussion

Cancer metabolism is rewired to facilitate the survival and proliferation of malignant tumor cells under diverse stress conditions[2,22]. Rapid cell proliferation consumes plenty of glucose, thereby causing severe glucose insufficiency and subsequently energy stress in the tumor microenvironment. This signaling reprograms the metabolic pathways to allow GBM cells to utilize other available energetic substrates. Herein, we reveal a metabolic switch from glycolysis to fructolysis in GBM cells living in a glucose deprivation environment. Mechanistically, stress-sensing protein kinases PERK and GCN2 phosphorylate eIF2α in response to glucose deprivation, and the transcription factor ATF4, activated by eIF2α phosphorylation-mediated selective translation, induces the expression of fructolytic genes, leading to a fructolysis-dependent GBM growth. *ATF4* KO, disruption of ATF4 binding motifs in the promoter regions of fructolytic genes, or pharmacological blockage of fructose utilization significantly inhibits GBM growth (Fig. 6d). This study shows that ATF4-dependent fructolysis is a type of ISR, depending on which GBM cells obtain an energy supply to overcome metabolic stress.

Our findings demonstrate that fructolysis is induced by the stress of glucose deprivation in an ATF4-dependent manner. Given that glucose and fructose coexist in the blood stream under normal physiological condition, fructolysis is dispensable in normal tissues.

However, rapid tumor growth inevitably causes microenvironmental metabolic stress, resulting in the ability of tumor cells to switch their energy supply from glycolysis to fructolysis. In this case, blockage of fructose utilization would be only adverse for cancer cells, but not normal tissues, implying that fructose metabolism could be a druggable target for GBM treatment. In addition, fructose is a major component of the current diet; excessive consumption of fructose is associated with an increased risk of many cancers[23–25]. Consistently, our data indicate that fructose is an important metabolic fuel for the growth of GBM cells.

Fructose uptake is mediated mainly by fructose transporter GLUT5 and three other enzymes, KHK, ALDOB, and triokinase, all of which are required to metabolize fructose[26]. We demonstrate that the stress-induced transcription factor ATF4 simultaneously controls the expression of GLUT5 and ALDOB, which are two essential proteins of the fructolysis pathway, highlighting that ATF4 acts as a master transcription factor for fructose metabolism.

Our findings demonstrate that around 40%, 80%, and 95% of TJ46, GSC23, and U87 cells, respectively, exhibit ATF4-positive IHC staining in tumor tissues. Intriguingly, similar percentage of negative IHC staining of uH2B was observed in these tissues. These observations are reminiscent of a significant correlation between the ATF4-positive rate and the uH2B-negative rate in GBM tissues. Given that loss of uH2B may be used as a semiquantitative marker for glucose deprivation in tumor tissues[14], thus these results also imply that the extent of glucose deprivation in tumor tissues may depend on the type of GBM cell lines used for brain tumorigenesis.

In conclusion, our study defines a role of ATF4 in regulating fructolysis. It is notable that GBM switches its energy supply from glycolysis to fructolysis in response to glucose insufficiency. This unique fructose utilization feature provides a promising therapeutic opportunity for GBM.

## Methods

### Materials

Rabbit monoclonal antibodies recognizing Ki67 (#ab92742, 1:5000 for IHC) were obtained from Abcam. Mouse monoclonal antibody recognizing tubulin (sc-23948, 1:1000 for immunoblotting) was purchased from Santa Cruz Biotechnology. Mouse monoclonal antibody against Flag tag (#F3165, 1:2000 for immunoblotting), rabbit polyclonal antibody recognizing KHK (#HPA007040, 1:1000 for immunoblotting), 2-Deoxy-D-glucose (2-DG) (#R004060), ethylenediaminetetraacetic acid (EDTA) (#E6758), phenylmethanesulfonyl fluoride (PMSF) (#P7626), PEI transfection reagents (#49553-93-7) and crystal violet (#C0775) were purchased from Sigma. Rabbit monoclonal antibody recognizing ATF4 (#11815, 1:100 for ChIP and 1:1000 for immunoblotting), phospho-eIF2α (Ser51) (#3398, 1:1000 for immunoblotting), eIF2α (#5324, 1:1000 for immunoblotting), phospho-ACC (Ser79) (#11818, 1:1000 for immunoblotting), ACC (#3676, 1:1000 for immunoblotting), cleaved PARP (#5625, 1:100 for IHC), ubiquityl-Histone H2B (Lys120) (#5546, 1:1000 for immunoblotting and 1:100 for IHC), Histone H2B (#12364, 1:1000 for immunoblotting) and SimpleChIP Enzymatic Chromatin IP Kit (Magnetic Beads) (#9003) were purchased from Cell Signaling Technology. Rabbit polyclonal antibody recognizing ALDOB (#18065-1-AP, 1:1000 for immunoblotting and 1:100 for IHC) were obtained from Proteintech. Rabbit polyclonal antibody recognizing GLUT5 (#PA580023, 1:100 for IHC), horseradish peroxidase (HRP)-conjugated goat anti-mouse and anti-rabbit secondary antibodies were obtained from Thermo Fisher Scientific. 2,5-anhydro-D-mannitol (2,5-AM) (#21673) were purchased from Cayman Chemical. GSK2656157 (#HY-13820) were obtained from MedChemExpress. GCN2-IN-2 (A-92) (#GC32771-5) and Compound C (#GC17243) were obtained from GLPBIO. D-[5-³H] fructose was purchased from PerkinElmer. Mouse chow (#120101) was purchased from SPF (Beijing) Biotechnology.

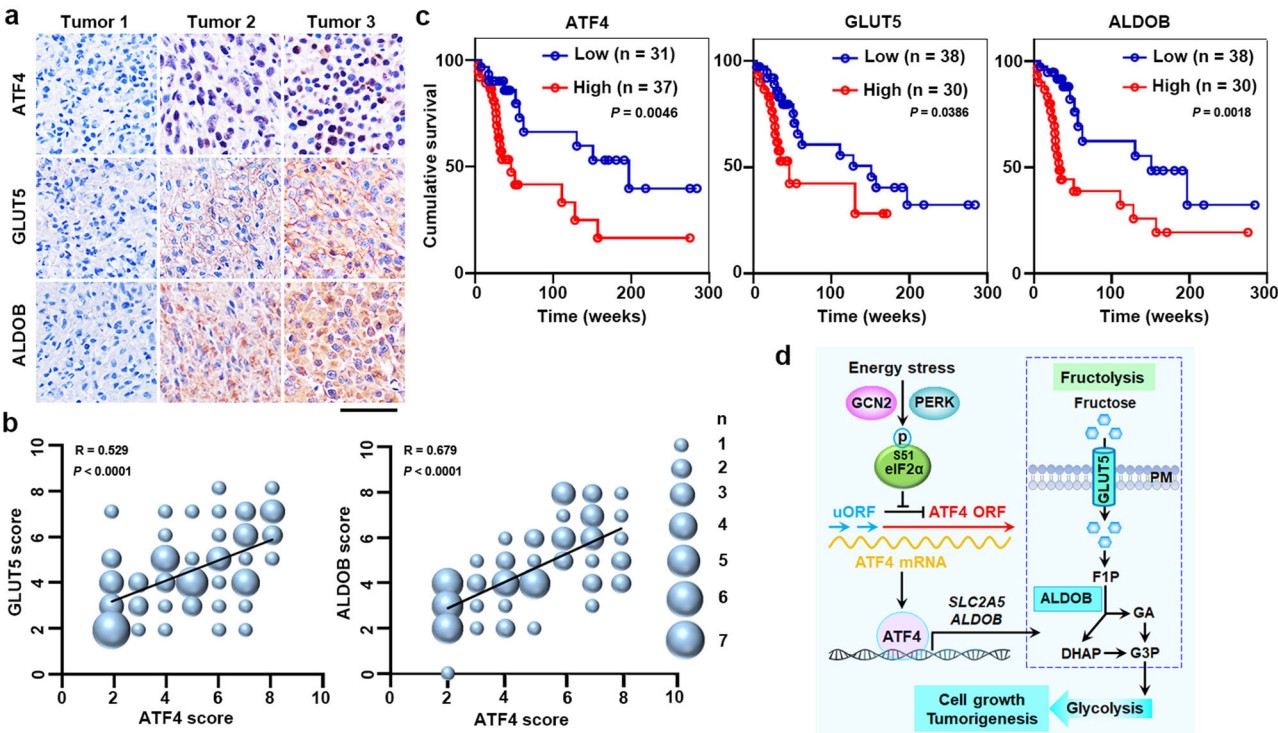

**Fig. 6 | The expression levels of ATF4, GLUT5, and ALDOB positively correlate with each other in GBM specimens and indicate a poor prognosis in GBM patients. a, b** Sixty-eight human primary GBM specimens were immunohistochemically stained with indicated antibodies. Representative photos of tumors are shown (**a**). Immunohistochemistry staining scores of ATF4, GLUT5, and ALDOB were analyzed by Spearman's correlation (**b**). Note that size (ranging from 1 to 7) of the dots represents the number of specimen(s) appearing on the graphs. Scale bar, 50 μm (**a**). **c** The survival time for 68 patients with low (1–4 staining scores, blue curve) versus high (5–8 staining scores, red curve) levels of ATF4, GLUT5, and ALDOB were compared. Empty circles represent censored data from patients alive at last clinical follow-up. *P* values were determined by the two-tailed log-rank test. **d** A mechanism of ATF4-dependent fructolysis supports GBM growth under glucose-deprived condition. A model shows that ATF4, which is activated upon glucose deprivation, induces expression of GLUT5 and ALDOB. High levels of GLUT5 and ALDOB drive fructolysis to maintain GBM growth under glucose-deprived condition. F1P fructose 1-phosphate, GA glyceraldehyde, DHAP dihydroxyacetone phosphate, GAP glyceraldehyde 3-phosphate. Source data are provided as a Source Data file.

## Cell lines and cell culture conditions

Human GBM cell lines including U87 (#HTB-14), LN229 (#CRL-2611), and A172 (#CRL-1620) were obtained from ATCC and maintained in Dulbecco's modified Eagle's medium (DMEM) supplemented with 10% dialyzed fetal bovine serum (HyClone) and 1% penicillin-streptomycin. Human GBM stem cell line GSC23 was maintained in neurobasal medium (#A2477501, Gibco) supplemented with B27, L-glutamine, sodium pyruvate, basic FGF (20 ng$^{-1}$ ml), EGF (20 ng$^{-1}$ ml) and different concentrations of glucose. Unless stated, the glucose-deprived media was made by supplementing the glucose-free DMEM (#11966025, Gibco) or neurobasal medium (#A2477501, Gibco) with 1 mM glucose. Unless stated, the media used for cell culture contains 25 mM glucose. The cell lines were routinely tested for mycoplasma contamination and authenticated by Short Tandem Repeat (STR) profiling.

The PDX cells derived from a GBM primary tissue were a gift from Drs. Chunsheng Kang and Chuan Fang (Tianjin Medical University) and used in previous publications[27,28]. The primary tumor tissue was surgically removed from a 46-year-old Chinese male GBM patient. Informed consent was obtained from the patient prior to the specimen collection. The use of patient tumor tissue and related database was approved by the Human Research Ethics Committee of Tianjin Medical University.

## RNA isolation and quantitative PCR

One microgram of total RNA extracted from cultured cells using the TRIzol reagent (Invitrogen) was reverse-transcribed into complementary DNA (cDNA) in a 20 μl reaction using the HiScript II Q RT Super Mix (Takara). TransStart Green qPCR SuperMix (TransGen Biotech) was used to determine the threshold cycle (Ct) value of each sample in a 20 μl reaction using the ABI Q7 Fast Real Time PCR System according to the manufacturer's instructions. The β-actin mRNA (*ACTB*) served as the normalization gene in these studies. ΔCt was defined as the Ct of *ACTB* minus the Ct of the target gene and the relative expression levels for the target genes were determined by $2^{\Delta Ct}$. Primer sequences used for PCR are listed in Supplementary Table 3.

## Immunoblotting

Total proteins of the cells cultured on 60-mm dishes were extracted by lysis buffer (50 mM Tris-HCl [pH 7.5], 0.1% SDS, 1% Triton X-100, 150 mM NaCl, 1 mM dithiothreitol, 0.5 mM EDTA, 100 μM PMSF, and 100 μM leupeptin). Total cell lysates were centrifuged at $13,400 \times g$ for 10 min at 4 °C to remove cell debris. Equal volume (20 μl) of supernatants (2 mg protein ml$^{-1}$) was mixed with 5× loading buffer (250 mM Tris-HCl [pH 6.8], 10% SDS, 50% glycerol, 0.5% bromophenol blue, and 5% β-mercaptoethanol) and heated for 5 min at 99 °C. The samples were separated by sodium dodecyl sulfate-polyacrylamide gel electrophoresis (SDS-PAGE) and transferred onto a PVDF membrane (GE Healthcare Life Sciences, PA) by wet or semi-dry transfer. The membranes were incubated with primary and then HRP-conjugated secondary antibodies. Immunoblots were visualized by a ChemiScope 6000 Exp instrument (CLinX, China) using the SuperSignal West Pico Chemiluminescent Substrate (Thermo Fisher Scientific).

## Generation of stable knockout cells

Guide RNAs (gRNAs) targeting the amino acid coding sequence of human *ATF4* were cloned into the lentiCRISPRv2 lentiviral vector

(#52961, Addgene) with selectable marker for puromycin. The empty vector lentiCRISPRv2 served as a control. Recombinant lentiviral particles were produced by co-transfection of lentiCRISPRv2 plasmid containing expression cassettes of hSpCas9 and the chimeric guide RNA, and two packaging plasmids pMD2.G (#12259, Addgene) and psPAX2 (#12260, Addgene) into 293FT cells in a 10-cm dish. Forty-eight hours after transfection, viral supernatant was collected for centrifugation at $180 \times g$ for 10 min at 4 °C to remove cell debris, and then filtered through 0.45 μm filter (Millipore). GBM cells pretreated with 5 μg ml$^{-1}$ of polybrene were infected with lentiviruses at a multiplicity of infection [MOI] of 1. Uninfected cells were removed by culturing with 1 μg ml$^{-1}$ of puromycin for 7 days and knockout efficiency was evaluated by immunoblotting. The detailed information of the gRNA nucleotide sequences targeting indicated genes are listed in Supplementary Table 3.

## Genomic editing

Genomic mutations in the promoters of *SLC2A5* and *ALDOB* were created using the prime editing (PE) system as described previously[29]. Briefly, GBM cells seeded onto a six-well plate at confluency of ~60% were co-transfected with pCMV-PE2-P2A-GFP plasmid (#132776, Addgene) and pU6-pegRNA-GG-acceptor plasmid containing the desired pegRNA. Three days after transfection, fluorescence-positive single cells were seeded into individual wells of 96-well plates by cell sorting. The sorted cells were grown for 4 weeks and then aliquoted for genomic DNA extraction. Genotyping was performed by sequencing the PCR products that were amplified using primers spanning the mutations created by the PE system. The cell clones carrying mutated CARE motifs (5′-TGATGGAAC-3′ to 5′-CACTAGGGC-3′ for *SLC2A5*; 5′-TGAGGAAAC-3′ to 5′-CACGAAGGC-3′ for *ALDOB*) in the promoters of *SLC2A5* or *ALDOB* were collected for further studies. The primer sequences used for PCR amplifications are listed in Supplementary Table 3.

## DNA constructs and mutagenesis

To construct an ATF4 exogenous expression plasmid, a PCR-amplified open reading frame DNA sequence of human *ATF4* was subcloned into lentiviral vector pCDH (#72265, Addgene) with hygromycin resistance. To construct the luciferase reporters, expression of which was driven by the promoter of *SLC2A5* or *ALDOB*, the PCR-amplified promoter region of *SLC2A5* or *ALDOB* was subcloned into lentiviral vector pGreenFire1 (#TR010PA-P, System Biosciences) with puromycin resistance. pGreenFire1-*SLC2A5* and pGreenFire1-*ALDOB* carrying mutated CARE motifs (5′-TGATG-GAAC-3′ to 5′-CACTAGGGC-3′ for *SLC2A5*; 5′-TGAGGAAAC-3′ to 5′-CACGAAGGC-3′ for *ALDOB*) were constructed using a QuikChange site-directed mutagenesis kit (Stratagene). To construct the pU6-pegRNA-GG-acceptor plasmid expressing the desired pegRNAs, annealed and phosphorylated pegRNA backbone oligonucleotides were ligated into pU6-pegRNA-GG-vector plasmid (#132777, Addgene) digested with BsaI-HFv2 restriction enzyme (#R3733, NEB). The oligonucleotide sequences used for construction of plasmids are listed in Supplementary Table 3.

## Cell proliferation and colony formation assays

For cell proliferation, a total of $1 \times 10^5$ cells suspended in 2 ml of medium were seeded into a six-well plate and maintained in DMEM supplemented with dialyzed 10% fetal bovine serum. The cells of each well were resuspended by trypsinization and counted daily. For colony formation, a total of 800 cells were seeded into a 12-well plate and maintained in DMEM supplemented with dialyzed 10% fetal bovine serum for 14 days. The cell colonies were photographed after staining with 2% crystal violet for 30 min following fixation with 4% paraformaldehyde for 20 min. Medium was changed daily for the cells used in proliferation and colony formation assays.

## Cell viability assay

GBM cells were plated at a density of $2 \times 10^5$ cells per well into the 6-well plates 12 h prior to severe glucose deprivation (a treatment with 0 mM glucose). The viable cells were counted using the staining of trypan blue (0.5%) at 24 h post severe glucose deprivation.

## Fructose metabolic rate assay

One million cells pretreated with or without glucose deprivation were washed once in PBS before incubation in 2 ml of Krebs buffer (126 mM NaCl, 2.5 mM KCl, 25 mM NaHCO$_3$, 1.2 mM NaH$_2$PO$_4$, 1.2 mM MgCl$_2$, and 2.5 mM CaCl$_2$) without fructose at 37 °C for 10 min. The Krebs buffer was then replaced with 2 ml of Krebs buffer supplemented with 5 mM fructose spiked with 10 μCi of D-[5-$^3$H] fructose. Following incubation of the cells for 1 h at 37 °C, triplicate 50 μl aliquots of the Krebs buffer were collected and mixed with 50 μl of HCl (0.2 N) in uncapped PCR tubes. The diffusion of $^3$H$_2$O was analyzed by transferring the uncapped PCR tube into an Eppendorf tube containing 0.5 ml of label free water as a recipient. The Eppendorf tubes were sealed and $^3$H$_2$O diffusing was kept for a minimum of 24 h at 37 °C. The $^3$H$_2$O diffusing to the recipient was detected using scintillation counting and normalized to cell number (per $1 \times 10^6$).

## Measurements of fructose consumption and serum fructose concentrations

A total of $1 \times 10^6$ cells suspended in 3 ml of medium were seeded into 60-mm dishes, 6 h later, the cells were incubated with medium supplemented with dialyzed serum in the presence or absence of glucose (25 mM) for 18 h, and then the cells were cultured for another 18 h following addition of fructose (10 mM). The media were collected for analysis of fructose consumption. The culture medium was then collected for measurement of the fructose concentration. The fructose levels in culture medium or mouse serum were determined using a fructose assay kit (#EFRU-100, BioAssay Systems) according to the manufacturer's instructions. Cells were collected and counted, and fructose consumption was normalized according to cell number (per $1 \times 10^6$).

## Luciferase reporter assay

The promoter regions ($-593$ to $+255$ bp from the TSS) of *SLC2A5* and ($-661$ to $+214$ bp from the TSS) of *ALDOB* were amplified by PCR and inserted into pGreenFire1 lentivector (System Biosciences). Lentiviruses were produced by co-transfecting 293FT cells with pGreenFire1 plasmid containing promoters of *SLC2A5* or *ALDOB*, and two packaging plasmids, pMD2.G (#12259, Addgene) and psPAX2 (#12260, Addgene). GBM cells were seeded into a 6-well plate and infected with the lentiviruses at a MOI of 1. The cell pools stably expressing luciferase under promoters of *SLC2A5* or *ALDOB* were generated by treating lentivirus-infected cells with 1 μg ml$^{-1}$ of puromycin for 7 days. To determine the ability of ATF4 to promote transcription, the cells were treated with or without glucose deprivation, and luciferase assays were performed using a Luciferase Assay System (Promega). Luciferase activities were normalized to the cell number obtained from the duplicated samples.

## ChIP assay

Chromatin immunoprecipitation (ChIP) assay was performed by using the SimpleChIP Enzymatic Chromatin IP kit (#9003, Cell Signaling Technology) according to the manufacturer's instructions. Chromatin prepared from GBM cells ($2 \times 10^7$) in a 15-cm dish was used to determine DNA input and incubate with anti-ATF4 antibody or control normal IgG overnight at 4 °C. The primer sequences used for PCR amplifications are listed in Supplementary Table 3.

## ChIP-seq analyses

Sequencing libraries were prepared using the NEBNext DNA Library Prep Master Mix Set (Illumina) according to the manufacturer's

instructions. The libraries were paired-end sequenced on the Illumina NovaSeq 6000 system (Annoroad Gene Technology, Beijing). Sequencing reads were qualified using FASTQC (v0.11.9) followed by adapter removal and reserving the reads with a length of at least 36 bases using the Trim Galore program (v0.6.7). The processed reads were aligned to GRCh38 (hg38) reference genome using BWA (v0.7.9a) followed by duplicate reads removal using sambamba (v0.8.0). The peaks were called from alignment results using the call peak function of MACS2 (v2.2.7.1) with default parameters and annotated by the annotatePeaks.pl function of HOMER (v4.8.3) using GENCODE19 as reference. The annotated peaks were filtered using the following criteria: each peak must have the "TSS" annotation and be located within the canonical chromosomes with false discovery rate (FDR) ≤ 0.1. The rest of the peaks were further filtered according to their distances (±1 kb) to the TSS.

The motifs present in the surrounding region of the peaks were searched by the findMotifsGenome.pl program of HOMER using the -S 5 option and the search region was constrained within ±50 bp of a peak position using the -size 50 option.

Total annotated and known ATF4 target gene sets, extracted from ENSEMBL GRCh38 ($n = 60,671$) and the previous publication[30] ($n = 472$), respectively, and fructolytic gene set ($n = 4$), including *SLC2A5*, *ALDOB*, *KHK*, and *TKFC*, were used to determine whether the fructolytic and ATF4 target genes were significantly enriched in our defined ATF4 ChIP peaks via a hypergeometric test. The hypergeometric test was performed by using the R statistical program (R Core Team, 2013).

## Whole-genome sequencing

For the whole-genome sequencing (WGS), the genomic DNA of one million TJ46 cells was extracted with a DNeasy Blood & Tissue Kit (#51304, Qiagen). DNA sequencing (30× coverage for the whole genome) was performed using the Illumina NovaSeq 6000 system (Annoroad Gene Technology, Beijing). Sequencing reads were qualified using FASTQC (v0.11.9). Next, adapters and low-quality bases in reads were removed using the Trim Galore program (v0.6.7). Usable reads were aligned to the GRCh37 (hg19) reference genome by the BWA software (v0.7.9a) with default parameters. Alignments were processed and sorted by the SAMtools package (v0.1.19) and then used for the identification of genomic variants. Single-nucleotide polymorphisms (SNPs) were identified with the GATK 3.8 scripts. Of note, PCR duplicates were marked by the Picard script (v1.115) and re-aligned to the local sequences around indels using the GATK 3.8 scripts to reduce false-positive detection of SNPs. The copy number variations (CNVs) were detected by the Control-FREEC 11.5 package and further annotated by the ANNOVAR program (v20200608).

## Orthotropic mouse model of GBM

Luciferase-expressing U87, TJ46, and GSC23 cells ($5 × 10^5$) suspended in 5 μl of medium were intracranially injected into 6-week-old male athymic nude mice ($n = 5$) obtained from GemPharmatech (Nanjing, China). Mice were sacrificed at the indicated time points after tumor cell injection. The brain of each mouse was dissected and harvested, fixed in 4% formaldehyde, and embedded in paraffin. The samples were sectioned in 5 μm-thickness. Tumor formation and phenotype were evaluated by histologic analysis of hematoxylin and eosin (H&E)-stained sections. The duplicated groups of tumor-bearing mice ($n = 10$) were used to determine the survival time, which is the duration between the time points of intracranial tumor cell injection and appearance of clear morbidity signs in each mouse. To evaluate the effect of 2,5-anhydro-D-mannitol (2,5-AM) on GBM progression, 2,5-AM-based treatments were initiated 7 days after intracranial injection of tumor cells. 2,5-AM (75 or 200 mg kg⁻¹ in 200 μl of PBS) was delivered to mice ($n = 5$) via intraperitoneal administration daily for the

duration of the experiment. To evaluate the effect of fructose administration on GBM progression, three days after tumor-cell injection, 200 μl of fructose (1 g kg⁻¹) or vehicle (PBS) was delivered to the mice via orogastrical administration daily for 18 days.

Mice were maintained in a pathogen-free environment with the relative humidity at 50 to 65%, temperature at 23 ± 2 °C, and light/dark cycle of 12 h/12 h. The animals with 3–5 mice per cage were free to access food [#120101, SPF (Beijing) Biotechnology] and water and treated in accordance with the Guide for the Care and Use of Laboratory Animals published by the National Academy of Sciences and the National Institutes of Health. The animal use in this study was approved and the maximal tumor size/burden was permitted by the Institutional Animal Care and Use Committee (IACUC) of the Center for Animal Experiments of the Institute of Biophysics, Chinese Academy of Sciences. All the animal studies were performed without exceeding the maximal tumor size/burden.

## Bioluminescent imaging

D-luciferin (450 mg kg⁻¹, Cayman Chemical) dissolved in 250 μl of PBS was intraperitoneally injected into the neck region of each mouse. The mice were photographed 10–20 min after D-luciferin administration. The IVIS Lumina System coupled with the Living Image data-acquisition software program (Xenogen Corporation, Alameda, CA) was used to record and quantify the peak photon flux within a target region.

## Immunohistochemical staining

The IHC staining was performed using a VECTASTAIN ABC kit (Vector Laboratories, CA) according to the manufacturer's instructions. Sections of paraffin-embedded tumor tissue were stained with antibodies against ATF4, GLUT5, ALDOB, Ki67, cleaved PARP, ubiquityl-Histone H2B (uH2B) or nonspecific IgG as a negative control. The tissue sections from paraffin-embedded human GBM specimens were stained with antibodies against ATF4, GLUT5, ALDOB, or nonspecific IgG as a negative control. The staining score of each tissue section was determined by the percentage of positive cells and staining intensity as previously defined[31]. For human GBM specimens, the proportion scores of the staining were assigned on a scale of 0–5, 0 if no tumor cells exhibited positive staining, 1 if 0–1%, 2 if 2–10%, 3 if 11–30%, 4 if 31–70%, and 5 if 71–100% and the intensity scores of the staining were assigned on a scale of 0–3, 0, negative; 1, weak; 2, moderate; and 3, strong. The proportion and intensity scores were then combined to generate a total score ranging from 0 to 8. We defined the staining scores of 0 to 4 as low staining, and 5 to 8 as high staining. The duration from the date of diagnosis to death, or last known date of follow-up, was defined as overall survival time of the patients. All patients had received standard therapy. All tissue samples were collected in compliance with informed consent policy and the use of patient tumor specimens, and the relevant database was approved by the Human Research Ethics Committee of the First Affiliated Hospital of Nanjing Medical University.

## Statistics and reproducibility

IBM SPSS Statistics software and R statistical program were used to perform statistical analyses. For paired comparison and multiple comparisons of variables, $P$ values were calculated by using two-tailed Student's $t$-test and one-way ANOVA, respectively. Survival rate comparison in the GBM patients and tumor-bearing mice was performed using the two-tailed log-rank test. The correlation between ATF4, GLUT5, and ALDOB levels was determined by the Spearman's coefficient. Multivariate analyses of GBM patient survival were performed using the Cox regression model. The statistical significance of ChIP-seq peak calls was determined by the two-tailed fisher's exact test. The enrichment of genes to fructolysis and ATF4 pathways was determined

by the two-tailed hypergeometric test. *P* values <0.05 were considered significant.

**Reporting summary**

Further information on research design is available in the Nature Research Reporting Summary linked to this article.

## Data availability

The ChIP-seq data (https://www.ncbi.nlm.nih.gov/geo/query/acc.cgi?acc=GSE188633) and TJ46 cell whole-genome sequencing data (https://www.ncbi.nlm.nih.gov/geo/query/acc.cgi?acc=GSE202644) that support the findings of this study have been deposited in the Gene Expression Omnibus. The data supporting the findings of this study are available within the article and its Supplementary Information file. Source data are provided with this paper.

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

## Acknowledgements

We thank Zhimin Wang (CAS Key Laboratory of Infection and Immunity, Institute of Biophysics) for the assistances in bioinformatic analyses and Hongjie Zhang (Core Facility of Protein Sciences, Institute of Biophysics) for technical support. This work was supported by the Key Program of the Chinese Academy of Sciences (Grant No. KJZD-SW-L05 to X.L.), the National Natural Science Foundation of China (Grant No. 82073060 to X.L.), the Training Program of the Major Research Plan of the National Natural Science Foundation of China (Grant No. 92157104 to X.L.), the National Key R&D Program of China (Grant No. 2020YFC2002700 to X.L.), and the National Science Foundation for Young Scientists of China (Grant No. 82103349 to C.C., and Grant No. 82003032 to Z.Z.).

## Author contributions

This study was conceived by X.L.; X.L. and C.C. designed the study; C.C., Z.Z., C.L., B.W., P.L., S.F., and F.Y. performed the experiments; Y.Y. provided reagents and pathology assistance; X.L. and C.C. wrote the manuscript with comments from all authors.

## Competing interests

The authors declare no competing interests.
