## [Peer Review File · Nature Communications]

REVIEWER COMMENTS

Reviewer #1 (Remarks to the Author):

The manuscript by Chen et al. is well-written, easy to follow, and provides new insights on fructose metabolism in glioblastoma (GBM) pathologies through an ATF4-dependent manner. The authors reported that fructose metabolism is upregulated upon glucose deprivation, thus sustaining the proliferation of cancer cells in vitro and in vivo. They showed this phenotype is ATF4-dependent by multiple knockout experiments, ChIP-seq, and reporter assays. They also demonstrated the utilities of fructose transporter inhibitor in treating GBM and showed ATF4 and fructolytic genes expression levels are prognostic markers in GBM patients. Although inhibition of GLUT5 for treating GBM has been reported (as cited in ref 4 by the author), hence, the usage of fructose in GBM is known, this article provides important insights on GBM metabolism adaptation in the microenvironment, putting enhanced ATF4-dependent fructose metabolism as a novel mechanism of cancer cell survival in responses to the reduced glucose source. Specific comments are listed below.

Major comments:

1. It is unknown the physiological relevance of using "glucose-deprived DMEM". Many studies used low glucose medium (e.g. 1.1 mM or 5 mM) and detailed titration of glucose concentration is needed to demonstrate the switch of fructose metabolism is not artificially induced by the complete absence of glucose/ fuel.
2. Related to point 1. The formulation (and source) of "glucose-deprived DMEM" was not provided. Does the medium include pyruvate or other potential fuels?
3. It has been shown that some GBM cell lines are resistant to glucose deprivation, such as A172 cells. It needs more justification for dropping this cell line in the rest of the figures after Figure 1. (PMID: 31914417)
4. The representation of Fig. 2A can be misleading as supplementary source data showed the fold changes were deduced by setting the untreated glucose-supplemented group as control. It is also unknown whether these inhibitors would induce mRNA expression changes in the presence of glucose. Similar concerns also apply to Fig. 2B.
5. It is currently unknown the level of ATF4 binding to the TSS and promoter regions for ALDOB and GLUT5 from the current data (Fig. 3A) when glucose is present.
6. Inconsistencies were found for Fig.3b legend and the corresponding data source that figure legend mentioned the cells were treated with or without glucose for 18 hours while labels in the data source were ATF4 OE. If it is treated with or without glucose, justification is needed for using luciferase assay which is ATP dependent.
7. More justification is needed for the choice of using MTT assay to measure cell proliferation as the assay highly depends on the metabolism of the cell while the deprivation of glucose/ other fuel will obviously shift the cell metabolism rate.
8. Using high glucose DMEM (25 mM) that might saturate the cell proliferation rate thus hindering assessment of the potential toxicity of the KO constructs in the presence of glucose (Supplementary Fig. 41-d).
9. It is now unclear when will the cancer start depending on the fructose (initiation v.s. progression when the tumor is large). Only IVIS image at day 14 was given in Fig. 5, thus whether the tumor sizes were different among groups at day 7 (pre-treatment). Similar concerns also apply to Fig. 4e-f where the date of IVIS was not mentioned.
10. Although it might be a sensible guess, there is no evidence (data or reference article) provided in the text there is depletion of glucose in GBM microenvironment.
11. The author should discuss the relative importance of fructose metabolism in GBM given that only around 40% of cells are positive with ATF4 (Supplementary Fig. 5c).

Minor comments:

12. Official name of ATF4 was never mentioned in the text.
13. The author should consider also including the qPCR data for KHK and TKFC after ATF4 KO/ OE to show the necessity of ATF4 binding for the observed up-regulation of mRNA expression of fructolytic genes.
14. More justification is needed for the choice of measuring cell proliferation rate at day 3 after glucose deprivation. A time-series cell proliferation/ counting assay should be performed.

15. It is unclear why there is an inconsistency of cell proliferation rate and colony area quantification between the cell line such that LN229 has a lower cell proliferation rate but more colony area coverage (Fig. 41-d). This trend was also maintained in the presence of glucose (Supplementary Fig. 41-d).
16. Color is not matching for Fig. 5e.
17. The author should justify the use of the Pearson correlation coefficient for calculating correlation of ordinal data rather than using the Spearman correlation coefficient.

Reviewer #2 (Remarks to the Author):

Chen and colleagues present a compelling study on the mechanism of fructolysis activation in GBM cells under conditions of glucose deprivation. Although prior studies have shown that GBM cells engage fructose metabolism to compensate for glucose deprivation, the precise molecular pathways that mediate this response have not been elucidated. The data presented in this study clearly show that ISR-dependent ATF4 activation promotes expression of GLUT5 and ALDOB, proteins that facilitate uptake and metabolism of fructose, respectively, during glucose starvation. The authors blend in vitro and in vivo work to reach this conclusion and identify fructose metabolism as a factor associated with GBM aggressiveness. Their use of CRISPR/Cas9 genome editing to abolish ATF4 binding sites in SLC2A5 and ALDOB gene regulatory loci is particularly effective in delineating the pathway linking glucose deprivation with fructolysis. This study makes an important contribution to knowledge regarding how brain tumor cells utilize alternative metabolic substrates and may provide insights into the molecular mechanisms of fructolysis activation in other disease contexts. I have relatively minor feedback overall.

Minor Points:

- 1) The information and nomenclature for the GBM "PDX" line are not optimal. Please provide information on the genetic alterations (key mutations and copy number alterations) that characterize this model. This information is needed to ensure that future studies can assess potential correlations between GBM oncogenotypes and fructolysis dependence. Also, in figures in which this model is used in vitro, I recommend not using the "PDX" label. The suggestion that "xenografts" (implied by use of the term PDX) are used in these experiments is not accurate and may be misleading to readers.
- 2) In Figure 6B, it appears that some data points represent multiple samples whereas others only represent one sample. Please use color coding or data point size to indicate the number of samples represented by each point.
- 3) The data in Supplemental Figures 4G, 4H, and 5D are very important for interpreting data in Figures 4 and 5. Without these data, there is no evidence that ATF4 loss or 2,5-AM treatment affects fructose metabolism in vivo. The data in Supplemental Figure 5D also indicate that GBM fructose consumption affects fructose levels in systemic circulation, which indicates high tumor fructose uptake. I recommend moving these panels to the main figures rather than including them in supplementary data.
- 4) Lines 251-252 in the Results section suggest a causal relationship between fructose metabolism and "clinical aggressiveness of human GBM". Data in Figure 6C can only support conclusions regarding correlation, not causation. Therefore, I recommend revising this sentence to indicate that ATF4-dependent fructolysis is associated with clinical aggressiveness of human GBM.
- 5) Lines 280-281 in the Discussion section state, "Therefore, a fructose free diet is recommended to GBM patients." This sentence should be removed because it is highly speculative and not based on the results of an interventional clinical trial testing the impact of fructose limitation in human GBM patients.
- 6) Is published data available that report on the extent of glucose depletion that is observed in the tumor microenvironment of human GBMs? If so, this should be cited and commented upon in the Discussion section. Given that fructolysis is dispensable under glucose-replete conditions, glucose depletion in the GBM microenvironment is highly relevant for the translational implications of this study.

Reviewer #3 (Remarks to the Author):

In this study Chen et al explored the role of fructolysis in GBM cancer proliferation and survival during glycolysis and tumorigenesis. They found that glucose deprivation induced ATF4 induction, which is required for fructolysis and GBM survival under glucose deprivation. Mechanistically, they showed that ATF4 binds to CARE motif in metabolic genes including GLUT5 and aldolase B involved in fructose uptake and fructolysis using CHIP and CHIP-seq. Mutations in the promoter region for ATF4 binding sites in GLUT5 or Aldolase B or pharmacologically targeting GLUT5 impaired GBM survival and colony formation under glucose deprivation. In preclinical models, they showed that GBM cells with ATF4 loss or mutation in loss or mutations in the promoter region for ATF4 binding sites in GLUT5 or Aldolase B impaired tumorigenesis of GBM, similar to GLUT5 targeting. Finally, they showed that ATF4 is positively correlated with GLUT5 and Aldolase and their overexpression predicts poor survival outcome in GBM patients. Overall, the datasets are in a great quality supports the most conclusions. and the study provides the novel concept to advances the field. Below are the comments that are essential and can help strengthen the study.

1. GLUT5 knockdown or Aldolase knockdown GBM cells should be generated to assess whether they are required for fructose-mediated GBM survival under glucose deprivation.
2. Does restoration of GLUT5 and aldolase rescues fructose-mediated GBM survival under glucose deprivation and the impairment of tumorigenesis of ATF4 knockdown cells.
3. The study has not demonstrated that fructose-promoted tumorigenesis depends on GLUT5 and aldolase B.
4. GLUT5 inhibitor used in the study may have off-target effects. It will be important to evaluate whether loss of GLUT5 or loss of Aldolase B impairs the efficacy of GLUT5 inhibitor in cell survival and tumorigenesis of GBM cells in comparison with control knockdown cells.
5. The PDX models from GBM patients should be included in the in vivo tumorigenesis study with a GLUT5 inhibitor to strengthen the study, although the xenograft model from PDX derived cell model is used.

RESPONSE TO REVIEWERS' COMMENTS

Reviewer #1 (Remarks to the Author):

The manuscript by Chen et al. is well-written, easy to follow, and provides new insights on fructose metabolism in glioblastoma (GBM) pathologies through an ATF4-dependent manner. The authors reported that fructose metabolism is upregulated upon glucose deprivation, thus sustaining the proliferation of cancer cells in vitro and in vivo. They showed this phenotype is ATF4-dependent by multiple knockout experiments, ChIP-seq, and reporter assays. They also demonstrated the utilities of fructose transporter inhibitor in treating GBM and showed ATF4 and fructolytic genes expression levels are prognostic markers in GBM patients. Although inhibition of GLUT5 for treating GBM has been reported (as cited in ref 4 by the author), hence, the usage of fructose in GBM is known, this article provides important insights on GBM metabolism adaptation in the microenvironment, putting enhanced ATF4-dependent fructose metabolism as a novel mechanism of cancer cell survival in responses to the reduced glucose source. Specific comments are listed below.

Answer: We greatly appreciate the reviewer's insightful and positive comments, which are essential for the improvement of this manuscript.

Major comments:

1. It is unknown the physiological relevance of using "glucose-deprived DMEM". Many studies used low glucose medium (e.g. 1.1 mM or 5 mM) and detailed titration of glucose concentration is needed to demonstrate the switch of fructose metabolism is not artificially induced by the complete absence of glucose/ fuel.

Answer: The reviewer's point is well taken. To titrate extracellular glucose concentration needed to activate fructose metabolism in GBM cells, we treated U87 and LN229 cells with media containing different glucose concentrations (0.3, 0.6, 0.9, 1.2, 2.4, 4.8 mM) and found that medium glucose levels lower than 1.2 mM are able to significantly upregulate fructolytic rate (Supplementary Fig. 1e) and induce expression of GLUT5 and ALDOB (Supplementary Fig. 1f). Consistently, upregulation of fructolytic rate (Supplementary Fig. 1g) and induced expression of GLUT5 and ALDOB (Supplementary Fig. 1h) resulted from glucose deprivation was abrogated by treating U87 and LN229 cells with media containing glucose levels of more than 1.2 mM. These results suggest that fructose metabolism in GBM cells is activated when extracellular glucose concentration is less than 1.2 mM.

2. Related to point 1. The formulation (and source) of "glucose-deprived DMEM" was not provided. Does the medium include pyruvate or other

potential fuels?

Answer: Glucose-free DMEM was purchased from Thermo Scientific (Gibco, catalog number: 11966025). According to the formulation provided by manufacturer (<https://www.thermofisher.cn/cn/en/home/technical-resources/media-formulation.49.html>), pyruvate is not included in this medium. Unless state, the “glucose-deprived DMEM” was made by supplementing the glucose-free DMEM with 10% (v/v) dialyzed fetal bovine serum (FBS) and 1 mM glucose. Detail information related to the formulation and source of "glucose-deprived DMEM" has been added into the Method section of the revised manuscript.

3. It has been shown that some GBM cell lines are resistant to glucose deprivation, such as A172 cells. It needs more justification for dropping this cell line in the rest of the figures after Figure 1. (PMID: 31914417)

Answer: The reviewer’s point is well taken. To test the effect of glucose deprivation on viability of GBM cells, we treated the U87, LN229, A172, and TJ46 (primary GBM cells) cells with severe glucose deprivation (a treatment with 0 mM glucose) for 24 hours. Consistent to the previous report (PMID: 31914417), we observed that A172 cells, but not U87, LN229, and TJ46 cells, are resistant to cell death induced by severe glucose deprivation (Supplementary Fig. 4e).

Indeed, our data demonstrate that fructolytic gene expression and fructolysis are activated by glucose deprivation in A172 cells (Fig. 1a-d). To address the reviewer's concern, we used U87, LN229, and A172 cells, which have different sensitivity to glucose-deprivation-induced cell death, to investigate the effects of fructolysis on proliferation of GBM cells in the revised manuscript. The results obtained from cell growth assay and colony formation assay indicated that fructose supplementation eliminated the glucose-deprivation-induced inhibition of cell growth (Fig. 4a) and colony formation (Fig. 4c) in GBM cells, including A172 cells, and this effect was abrogated by *ATF4* KO (Fig. 4a, c). These results suggest that ATF4-dependent fructolysis also supports proliferation and growth of A172 cells under glucose-deprived condition, although these cells (A172) are resistant to cell death resulted from severe glucose deprivation (Supplementary Fig. 4e).

4. The representation of Fig. 2A can be misleading as supplementary source data showed the fold changes were deduced by setting the untreated glucose-supplemented group as control. It is also unknown whether these inhibitors would induce mRNA expression changes in the presence of glucose. Similar concerns also apply to Fig. 2B.

Answer: The reviewer's point is well taken. To eliminate the confusion, we repeated the experiment of Fig. 2a and presented the relative mRNA

levels of *SLC2A5* and *ALDOB* under both glucose-supplemented (Glc +) and -deprived (Glc -) condition (Fig. 2a). These data demonstrated that treatments of the inhibitors (GSK2656157 and A-92) altered the mRNA expression of *SLC2A5* and *ALDOB* under glucose-deprived condition, in contrast, these effects were not observed under glucose-supplemented condition (Fig. 2a). Likewise, we obtained the consistent results in the repeated experiment of Fig. 2b.

5. It is currently unknown the level of ATF4 binding to the TSS and promoter regions for *ALDOB* and *GLUT5* from the current data (Fig. 3A) when glucose is present.

Answer: We have performed the reviewer's suggested experiments and added the results (Fig. 3a) into the revised manuscript.

6. Inconsistencies were found for Fig.3b legend and the corresponding data source that figure legend mentioned the cells were treated with or without glucose for 18 hours while labels in the data source were ATF4 OE. If it is treated with or without glucose, justification is needed for using luciferase assay which is ATP dependent.

Answer: We apologize for the incorrect labeling in the source data. We repeated the experiments of previous Fig. 3b and obtained the consistent results (Fig. 3b). In addition, luciferase activity is measured by adding the

substrate reagents, including ATP and luciferin, to cell lysates containing luciferase. Therefore, the luciferase assay performed in Fig. 3b does not depend on intracellular ATP levels, which may be affected by glucose deprivation.

7. More justification is needed for the choice of using MTT assay to measure cell proliferation as the assay highly depends on the metabolism of the cell while the deprivation of glucose/ other fuel will obviously shift the cell metabolism rate.

Answer: The reviewer's point is well taken. To evaluate the cell proliferation using a method that is insensitive to cell metabolism status, we seeded the GBM cells into 6-well plates and counted the cell number at different time points (Fig. 4a, 4b, 5a and Supplementary Fig. 4a, 4b, 6a). As expected, we obtained the results consistent to MTT assays.

8. Using high glucose DMEM (25 mM) that might saturate the cell proliferation rate thus hindering assessment of the potential toxicity of the KO constructs in the presence of glucose (Supplementary Fig. 41-d).

Answer: The reviewer's point is well taken. We cultured the GBM cells with medium containing high (25 mM) or normal (6 mM) glucose in presence or absence of 10 mM of fructose. The cell number was counted at day 3 post cell seeding. As data indicated in Supplementary Fig. 4a, b,

cells cultured with medium containing high or normal glucose in absence or presence of 10 mM of fructose exhibit similar proliferation rate. These results suggest that blocking ATF4-dependent fructolysis impairs GBM cell proliferation under glucose-deprived condition, but not high (25 mM) or normal (6 mM) glucose condition.

9. It is now unclear when will the cancer start depending on the fructose (initiation v.s. progression when the tumor is large). Only IVIS image at day 14 was given in Fig. 5, thus whether the tumor sizes were different among groups at day 7 (pre-treatment). Similar concerns also apply to Fig. 4e-f where the date of IVIS was not mentioned.

Answer: We have performed the reviewer's suggested experiments. We have visualized the tumor via bioluminescent imaging at day 7 post tumor-cell injection, which is the time point starting 2,5-AM (GLUT5 inhibitor) treatment, finding that there is no significant difference between the tumor size of each group (Fig. 5c). In addition, we measured tumor-derived luminescence intensity on day 3 and day 21 post tumor-cell injection (Fig. 4e, f) and obtained the results showing that blocking ATF4-dependent fructolysis does not alter the size of tumors at early stage (day 3 post tumor-cell injection), in contrast, tumors with ATF4-dependent fructolysis deficiency exhibited a significant decrease of tumor size at late stage (day 21 post tumor-cell injection). These data suggest that ATF4-dependent

fructolysis is instrumental during tumor progression, but not tumor initiation.

10. Although it might be a sensible guess, there is no evidence (data or reference article) provided in the text there is depletion of glucose in GBM microenvironment.

Answer: The reviewer's point is well taken. To address the reviewer's concern, we treated U87 and LN229 cells with media containing different concentrations of glucose. Immunoblotting analyses demonstrated that uH2B (K120 mono-ubiquitination of histone H2B), a semiquantitative histone marker for examining glucose levels that the tumor cells are exposed to (PMID: 22615809), was undetectable in U87 and LN229 cells treated with media containing glucose levels of lower than 1.2 mM (Supplementary Fig. 1f, h), suggesting that loss of uH2B may be used as a marker for glucose deprivation in tumor tissues. Next, to evaluate the glucose deprivation in GBM microenvironment, tumor tissues derived from U87 cells were collected on day 21 post tumor-cell injection. Immunohistochemical analyses of uH2B demonstrated that around 95% of tumor cells exhibited negative staining of uH2B on the tumor sections of each group (Supplementary Fig. 5e, f), evidencing that a large portion (~95%) of U87 cells in the tumor microenvironment underwent glucose deprivation. Basing on the similar strategy, we evaluated that around 40%

of TJ46 cells in the tumor microenvironment undergoes glucose deprivation (Supplementary Fig. 6f).

11. The author should discuss the relative importance of fructose metabolism in GBM given that only around 40% of cells are positive with ATF4 (Supplementary Fig. 5c).

Answer: A yellow-highlighted paragraph discussing the relative importance of fructose metabolism in GBM has been added into the revised manuscript.

Minor comments:

12. Official name of ATF4 was never mentioned in the text.

Answer: ATF4 official name, activating transcription factor 4, has been shown in the Abstract and main text of revised manuscript.

13. The author should consider also including the qPCR data for *KHK* and *TKFC* after ATF4 KO/ OE to show the necessity of ATF4 binding for the observed up-regulation of mRNA expression of fructolytic genes.

Answer: We have performed the reviewer's suggested experiments. Consistent to Fig. 1c, we observed that glucose deprivation did not alter mRNA expression levels of *KHK* and *TKFC* in wild type and ATF4-knockout GBM cells (Fig. 2d). Likewise, ATF4-overexpressing did not

alter mRNA expression levels of *KHK* and *TKFC* in GBM cells (Supplementary Fig. 2b). These data support that *KHK* and *TKFC* are expressed in GBM cells in an ATF4-independent manner.

14. More justification is needed for the choice of measuring cell proliferation rate at day 3 after glucose deprivation. A time-series cell proliferation/ counting assay should be performed.

Answer: We have performed the reviewer's suggested experiments. Results obtained from time-series cell proliferation/counting assays have been shown in Fig. 4a, b of the revised manuscript.

15. It is unclear why there is an inconsistency of cell proliferation rate and colony area quantification between the cell line such that LN229 has a lower cell proliferation rate but more colony area coverage (Fig. 41-d). This trend was also maintained in the presence of glucose (Supplementary Fig. 41-d).

Answer: This point is well taken. To avoid the inaccuracy caused by over confluence of cells, we seeded 2×10^3 cells suspended in 100 μ l of medium into a 96-well plate for cell proliferation assays. Both of U87 and LN229 cells did not exhibit over-confluence when cell proliferation assays were performed on day 3 post cell seeding. In contrast, these cells had exhibited over-confluence when colony formation assays were performed on day 14

post cell seeding. Therefore, the inconsistency of cell proliferation rate and colony area quantification between U87 and LN229 cells may be caused by their different sensitivity to confluence during cell growth.

16. Color is not matching for Fig. 5e.

Answer: We have corrected these mistakes in the revised manuscript.

17. The author should justify the use of the Pearson correlation coefficient for calculating correlation of ordinal data rather than using the Spearman correlation coefficient.

Answer: This point is well taken. Pearson's and Spearman's correlation assess linear and monotonic (whether linear or not) relationship, respectively, between two sets of data. Spearman's coefficient is appropriate for calculating correlation between immunohistochemistry staining scores, which are a kind of ordinal data. The correlations between immunohistochemistry staining scores of ATF4, GLUT5, and ALDOB have been analyzed by Spearman's coefficient in the revised manuscript (Fig. 6b).

Reviewer #2 (Remarks to the Author):

Chen and colleagues present a compelling study on the mechanism of

fructolysis activation in GBM cells under conditions of glucose deprivation. Although prior studies have shown that GBM cells engage fructose metabolism to compensate for glucose deprivation, the precise molecular pathways that mediate this response have not been elucidated. The data presented in this study clearly show that ISR-dependent ATF4 activation promotes expression of GLUT5 and ALDOB, proteins that facilitate uptake and metabolism of fructose, respectively, during glucose starvation. The authors blend in vitro and in vivo work to reach this conclusion and identify fructose metabolism as a factor associated with GBM aggressiveness. Their use of CRISPR/Cas9 genome editing to abolish ATF4 binding sites in SLC2A5 and ALDOB gene regulatory loci is particularly effective in delineating the pathway linking glucose deprivation with fructolysis. This study makes an important contribution to knowledge regarding how brain tumor cells utilize alternative metabolic substrates and may provide insights into the molecular mechanisms of fructolysis activation in other disease contexts. I have relatively minor feedback overall.

Answer: We greatly appreciate the reviewer's acknowledgement of the potential significance of this report and the insightful comments.

Minor Points:

1) The information and nomenclature for the GBM "PDX" line are not

optimal. Please provide information on the genetic alterations (key mutations and copy number alterations) that characterize this model. This information is needed to ensure that future studies can assess potential correlations between GBM oncogenotypes and fructolysis dependence. Also, in figures in which this model is used in vitro, I recommend not using the “PDX” label. The suggestion that “xenografts” (implied by use of the term PDX) are used in these experiments is not accurate and may be misleading to readers.

Answer: The reviewer’s point is reasonable. In the revised manuscript, the GBM “PDX” cell line has been changed its name to **TJ46**, since it is derived from a 46-year-old GBM patient receiving treatments in Tianjin, a city situated in northern China. Furthermore, the genetic alterations (key mutations and copy number alterations) existing in TJ46 cells have been investigated by the whole-genome sequencing of these cells (**Supplementary Table 1**). To eliminate the inaccuracy that may be misleading to readers, the term “xenografts” has been removed in the revised manuscript.

2) In Figure 6B, it appears that some data points represent multiple samples whereas others only represent one sample. Please use color coding or data point size to indicate the number of samples represented by each point.

Answer: This point is well taken. We have used the data point size to

indicate the number of samples represented by each point in Fig. 6b of the revised manuscript.

3) The data in Supplemental Figures 4G, 4H, and 5D are very important for interpreting data in Figures 4 and 5. Without these data, there is no evidence that ATF4 loss or 2,5-AM treatment affects fructose metabolism in vivo. The data in Supplemental Figure 5D also indicate that GBM fructose consumption affects fructose levels in systemic circulation, which indicates high tumor fructose uptake. I recommend moving these panels to the main figures rather than including them in supplementary data.

Answer: This point is well taken. We have followed the reviewer's suggestion to move previous Supplementary Figure 5d to the main figure as Fig. 5e.

4) Lines 251-252 in the Results section suggest a causal relationship between fructose metabolism and "clinical aggressiveness of human GBM". Data in Figure 6C can only support conclusions regarding correlation, not causation. Therefore, I recommend revising this sentence to indicate that ATF4-dependent fructolysis is associated with clinical aggressiveness of human GBM.

Answer: We have followed the reviewer's suggestion to change the original statement "Thus, these results suggest that ATF4-dependent

fructolysis plays a critical role in the clinical aggressiveness of human GBM.” in lines 251-252 to “Thus, these results indicate that ATF4-dependent fructolysis is associated with clinical aggressiveness of human GBM.” in the revised manuscript.

5) Lines 280-281 in the Discussion section state, “Therefore, a fructose free diet is recommended to GBM patients.” This sentence should be removed because it is highly speculative and not based on the results of an interventional clinical trial testing the impact of fructose limitation in human GBM patients.

Answer: We have followed the reviewer’s suggestion to remove the statement “Therefore, a fructose free diet is recommended to GBM patients.” in lines 280-281 of the Discussion section.

6) Is published data available that report on the extent of glucose depletion that is observed in the tumor microenvironment of human GBMs? If so, this should be cited and commented upon in the Discussion section. Given that fructolysis is dispensable under glucose-replete conditions, glucose depletion in the GBM microenvironment is highly relevant for the translational implications of this study.

Answer: The reviewer’s point is well taken. We failed to obtain the published data that report the extent of glucose deprivation observed in the

tumor microenvironment of human GBM. However, to address the reviewer's concern, we treated U87 and LN229 cells with media containing different concentrations of glucose. Immunoblotting analyses demonstrated that uH2B (K120 mono-ubiquitination of histone H2B), a semiquantitative histone marker for examining glucose levels that the tumor cells are exposed to (PMID: 22615809), was undetectable in U87 and LN229 cells treated with media containing glucose levels of lower than 1.2 mM (Supplementary Fig. 1f, h), suggesting that loss of uH2B may be used as a marker for glucose deprivation in tumor tissues. Next, to evaluate the glucose deprivation in GBM microenvironment, tumor tissues derived from U87 cells were collected on day 21 post tumor-cell injection. Immunohistochemical analyses of uH2B demonstrated that around 95% of tumor cells exhibited negative staining of uH2B on the tumor sections of each group (Supplementary Fig. 5e, f), evidencing that a large portion (~95%) of U87 cells in the tumor microenvironment underwent glucose deprivation. Basing on the similar strategy, we evaluated that around 40% of TJ46 cells in the tumor microenvironment undergoes glucose deprivation (Supplementary Fig. 6f).

Reviewer #3 (Remarks to the Author):

In this study Chen et al explored the role of fructolysis in GBM cancer

proliferation and survival during glycolysis and tumorigenesis. They found that glucose deprivation induced ATF4 induction, which is required for fructolysis and GBM survival under glucose deprivation. Mechanistically, they showed that ATF4 binds to CARE motif in metabolic genes including GLUT5 and aldolase B involved in fructose uptake and fructolysis using CHIP and CHIP-seq. Mutations in the promoter region for ATF4 binding sites in GLUT5 or Aldolase B or phenologically targeting GLUT5 impaired GBM survival and colony formation under glucose deprivation. In preclinical models, they showed that GBM cells with ATF4 loss or mutation in loss or mutations in the promoter region for ATF4 binding sites in GLUT5 or Aldolase B impaired tumorigenesis of GBM, similar to GLUT5 targeting. Finally, they showed that ATF4 is positively correlated with GLUT5 and Aldolase and their overexpression predicts poor survival outcome in GBM patients. Overall, the datasets are in a great quality supports the most conclusions. and the study provides the novel concept to advances the field. Below are the comments that are essential and can help strengthen the study.

Answer: We greatly appreciate the reviewer's insightful and positive comments, which significantly strengthened the manuscript.

1. GLUT5 knockdown or Aldolase knockdown GBM cells should be generated to assess whether they are required for fructose-mediated GBM

survival under glucose deprivation.

Answer: We have performed the reviewer's suggested experiments. We generated the *SLC2A5*- or *ALDOB*-knockdown U87 cells by infecting U87 cells with lentiviruses expressing shRNA targeting *SLC2A5* or *ALDOB* (Supplementary Fig. 4f). As expected, *SLC2A5* or *ALDOB* knockdown impaired the fructose-mediated colony formation of U87 cells under glucose-deprived condition (Supplementary Fig. 4g), suggesting that GLUT5 and ALDOB are required for fructose-mediated GBM cell survival under glucose-deprived condition.

2. Does restoration of GLUT5 and aldolase rescues fructose-mediated GBM survival under glucose deprivation and the impairment of tumorigenesis of ATF4 knockdown cells.

Answer: We have performed the reviewer's suggested experiments. We exogenously expressed the GLUT5 and ALDOB in *ATF4*-knockdown U87 cells (Supplementary Fig. 4h). A colony formation assay demonstrated that exogenous expression of GLUT5 and ALDOB rescued the inhibition of fructose-mediated colony formation in *ATF4*-knockdown U87 cells under glucose-deprived condition (Supplementary Fig. 4i). Similarly, orthotopic tumorigenesis analyses demonstrated that exogenous expression of GLUT5 and ALDOB eliminated the *ATF4*-knockdown-mediated impairment of GBM growth (Supplementary Fig. 5a).

3. The study has not demonstrated that fructose-promoted tumorigenesis depends on GLUT5 and aldolase B.

Answer: We orogastrically delivered fructose to nude mice bearing tumors derived from U87 cells without or with knockdown of *SLC2A5* or *ALDOB* (Supplementary Fig. 4f). As results indicated in Supplementary Fig. 5b, fructose administration promoted growth of tumors derived from U87 cells, but this effect was not observed in tumors derived from U87 cells with knockdown of *SLC2A5* or *ALDOB*, suggesting that fructose-promoted GBM growth depends on GLUT5 and ALDOB.

4. GLUT5 inhibitor used in the study may have off-target effects. It will be important to evaluate whether loss of GLUT5 or loss of Aldolase B impairs the efficacy of GLUT5 inhibitor in cell survival and tumorigenesis of GBM cells in comparison with control knockdown cells.

Answer: The reviewer's point is well taken. A colony formation assay demonstrated that 2,5-AM treatment impaired the fructose-mediated colony formation of U87 cells under glucose-deprived condition, but this effect was not observed in U87 cells with knockdown of *SLC2A5* or *ALDOB* (Supplementary Fig. 4f, 6c). Consistently, orthotopic tumorigenesis analyses demonstrated that 2,5-AM treatment inhibited growth of tumors derived from U87 cells, but this effect was not observed

in tumors derived from U87 cells with knockdown of *SLC2A5* or *ALDOB* (Supplementary Fig. 4f, 6d). These data support that 2,5-AM treatment inhibits GBM cell survival and GBM growth depending on GLUT5 and ALDOB expression.

5. The PDX models from GBM patients should be included in the in vivo tumorigenesis study with a GLUT5 inhibitor to strengthen the study, although the xenograft model from PDX derived cell model is used.

Answer: The reviewer's point is well taken. We have tried to perform the reviewer's suggested experiments. However, we failed to establish a GBM PDX mouse model due to the low rate of tumorigenesis derived from the GBM primary tissue samples. Alternatively, to strengthen our study, we evaluated the effect of 2,5-AM on tumor growth following the intracranial injection of the luciferase-expressing GBM stem cell line GSC23 established from the human GBM primary tissue into athymic nude mice. As expected, 2,5-AM administration significantly suppressed the growth of inoculated tumors in a dose-dependent manner (Fig. 5c), which was accompanied by a considerably prolonged survival time (Fig. 5d).

REVIEWERS' COMMENTS

Reviewer #2 (Remarks to the Author):

The authors have addressed all the points I raised. I recommend accepting the paper.

Reviewer #3 (Remarks to the Author):

The authors have adequately addressed all of my concerns. I have no more questions to ask. The study is novel and well supported by the experimental datasets. It is now suitable for publication at Nature Communications.

Reviewer #4 (Remarks to the Author):

The authors have appropriately addressed the former Reviewer 1's concerns with additional experiments as suggested and substantial amount of data. The revised manuscript has been improved and now suitable for publication in Nature Communications.